

# A thermodynamic framework for bulk–surface partitioning in finite-volume mixed organic–inorganic aerosol particles and cloud droplets

Ryan Schmedding[1] and Andreas Zuend[1]

[1]Department of Atmospheric and Oceanic Sciences, McGill University, Montreal, Quebec, Canada
**Correspondence:** Andreas Zuend (andreas.zuend@mcgill.ca)

**Abstract.** Atmospheric aerosol particles and their interactions with clouds are among the largest sources of uncertainty in global climate modeling. Aerosol particles in the ultrafine size range with diameters less than 100 nm have very high surface area to volume ratios, with a substantial fraction of molecules occupying the air–droplet interface. The partitioning of surface-active species between the interior bulk of a droplet and the interface with the surrounding air plays a large role in the

physicochemical properties of a particle and in the activation of ultrafine particles, especially those of less than 50 nm diameter, into cloud droplets. In this work, a novel and thermodynamically rigorous treatment of bulk–surface equilibrium partitioning is developed through the use of a framework based on the Aerosol Inorganic–Organic Mixtures Functional groups Activity Coefficients (AIOMFAC) model in combination with a finite-depth Guggenheim interface region on spherical, finite-volume droplets. We outline our numerical implementation of the resulting modified Butler equation, including accounting for chal-

lenging extreme cases when certain compounds have very limited solubility in either the surface or bulk phase. This model, which uses a single, physically constrained interface thickness parameter, is capable of predicting the size-dependent surface tension of complex multicomponent solutions containing organic and inorganic species. We explore the impacts of coupled surface tension changes and changes in bulk–surface partitioning coefficients for aerosol particles ranging in diameters from several μm to as small as 10 nm and across atmospherically relevant relative humidity ranges. The treatment of bulk–surface

equilibrium leads to deviations from classical cloud droplet activation behavior as modeled by simplified treatments of the Köhler equation that do not account for bulk–surface partitioning. The treatments for bulk–surface partitioning laid out in this work, when applied to the Köhler equation, are in agreement with measured critical supersaturations of a range of different systems. However, we also find that challenges remain in accurately modeling the growth behavior of certain systems containing small dicarboxylic acids, especially in a predictive manner. Furthermore, it was determined that the thickness of the interfacial

phase is sensitive parameter in this treatment; however, constraining it to a meaningful range allow for predictive modeling of aerosol particle activation into cloud droplets, including cases with consideration of co-condensation of semivolatile organics.

## 1 Introduction

Atmospheric aerosols are suspensions of condensed particles and the gas which surrounds them. They can be a major contributor to poor indoor and outdoor air quality. Exposure to poor air quality is the leading environmental risk factor for premature



mortality globally (Cohen et al., 2017; Burnett et al., 2018). Beyond their public health effects, the role of atmospheric aerosols and of aerosol–cloud interactions in the global climate system, remains one of the least understood processes in climate models (Boucher et al., 2013). Some light-absorbing aerosols exhibit a positive direct radiative forcing, while others mostly scatter solar radiation and exhibit negative direct radiative forcing. The magnitude and sign of the local or regional direct radiative forcing depends on numerous factors including an aerosol's composition, size distribution, geographic location, season, and altitude

in the atmosphere (Bellouin et al., 2020). Furthermore, atmospheric aerosols interact with clouds and precipitation, thereby contributing to important indirect effects on Earth's climate that are poorly constrained (Seinfeld et al., 2016). Therefore, it is of the utmost importance to understand the physicochemical properties and microphysical processes of aerosol particles, including aerosol–cloud–radiation interactions, as these drive their impacts on the climate system.

Aerosol particles exist over a broad range of sizes. Differently sized particles frequently originate from distinct sources or

form through unique processes. Particles in the nucleation mode are primarily formed via the spontaneous condensation of gaseous compounds to create or extend a tiny condensed phase. Similarly, Aitken mode particles may form by the continued growth of nucleation mode particles through condensation of gaseous species or coagulation of condensed particles. Particles in the nucleation mode or Aitken mode are collectively referred to as ultrafine particles. Accumulation mode particles form through coagulation and agglomeration of smaller Aitken mode particles in the air. The largest aerosols, such as mineral dust

and sea spray particles, are primarily composed of inorganic compounds and typically form via mechanical processes.

As aerosol particles decrease in size, they are more likely to originate from and grow through complex multiphase chemical reactions. Many fine and ultrafine aerosol particles are composed of organic compounds that form via oxidation reactions of volatile organic compounds (VOCs)or intermediate-volatility organic compounds (IVOCs) in the gas phase. Many of the products of these reactions are lower in volatility and partition into condensed particles. Semivolatile and low-volatility organic

compounds (SVOCs and LVOCs) may also be emitted and partition into particles with minimal processing in the gas phase. Organic aerosol particles, or components thereof, formed through gas-phase or multiphase chemistry are known as Secondary Organic Aerosols (SOA) (Jimenez et al., 2009; Hallquist et al., 2009); for a complete list of abbreviations and symbols, please refer to Tables S1 & S2 of the supplementary information (SI). Within a condensed phase additional chemical reactions may proceed. This is particularly notable in low-viscosity aqueous aerosol particles and cloud droplets, those droplets provide

aqueous media wherein numerous laboratory and field observations have observed the formation or transformation of SOA through chemical processing (e.g., Ervens et al., 2011). For certain species these reactions can further reduce their volatility, effectively "trapping" said species in the particle phase. For example, isoprene expoxydiols (IEPOX) are known to undergo acid-catalyzed reactions in aqueous $SO_4^{2-}$-containing particle phases (Surratt et al., 2010). The IVOCs and VOCs which lead to the formation of SOA can be biogenic (e.g. isoprene or monoterpenes) or anthropogenic (e.g. toluene, xylene, naphthalene)

Ng et al. (2007); Chan et al. (2009). Observational studies have concluded that the majority of SOA material is derived from biogenic precursors (Zhang et al., 2007).

The internal mixing state and geometries of aerosol particles are varied. Aerosol particles that are emitted by primary emission from biomass burning often take the shape of long agglomerations of individual spherical particles (Pokhrel et al., 2021), while aqueous inorganic-rich particles and soluble SOA are thought to be approximately spherical. The frequent presence of





liquid–liquid phase separation (LLPS) can complicate the shape aspect. LLPS particles may form radially symmetric structures with an organic-rich "shell" (phase $\beta$) covering an inorganic-rich "core" (phase $\alpha$) or other, geometrically more complex structures, such as a partially engulfed morphology, wherein the particle is no longer radially symmetric nor perfectly spherical. In a partially engulfed particle, phase $\beta$ does not spread completely over phase $\alpha$ and leaves a portion of phase $\alpha$ exposed (Song et al., 2013, 2012; Ciobanu et al., 2009; Shiraiwa et al., 2013). These particles are also called "russian dolls" or "janus particles" and interact with radiation differently than their core–shell counterparts (Lang-Yona et al., 2010). Furthermore, should phase $\beta$ of core–shell phase-separated particles become a highly viscous liquid or even glassy, it can limit the reactive uptake of different species (Schmedding et al., 2020, 2019; Zhou et al., 2019; Kuwata and Martin, 2012).

Since the surface area to volume ratio of a sphere scales with the inverse of its radius, said ratio becomes of great importance for ultrafine particles. If the assumption is made that the surface of a particle is represented by a monolayer of molecules in contact with the gas phase, an assumption further explored in this study, then spherical particles with a diameter less than 100 nm will have a non-negligible quantity of molecules present at their surface. For example, in a pure water droplet with a diameter of 50 nm, $\sim 4\%$ of the water molecules will be present in the surface monolayer while more than half of the water molecules in a pure droplet with a diameter of 3 nm will be present in the surface monolayer.

Many ultrafine aerosol particles are complex multicomponent systems and contain numerous surfactant and tensoionic species in differing concentrations. These species modify the surface tension of a particle (Sorjamaa et al., 2004; Nozière et al., 2014; Gérard et al., 2016; Petters and Petters, 2016; Ruehl et al., 2016; Ovadnevaite et al., 2017; Kroflič et al., 2018; Malila and Prisle, 2018; Gérard et al., 2019). In finite-volume systems, surfactants may become depleted out of the particle interior, likewise tensoionic species may become increasingly concentrated in the particle interior. This may influence the equilibrium morphology of an aerosol particle for a given size. Even in macroscopic systems, it is possible for LLPS particles to adopt non-spherical partially engulfed morphologies based on their surface properties (Binyaminov et al., 2021). The surface enrichment and depletion of species may affect the conditions under which an aerosol particle will activate and quickly grow into a cloud droplet.

The impact of surface tension modification by surfactants on cloud droplet activation has been known and studied for decades (Facchini et al., 1999, 2000; Topping et al., 2007; Ovadnevaite et al., 2017). The conditions under which a hygroscopically growing aerosol particle will activate into a cloud droplet were first stated by Köhler (1936), recognizing the importance of the global maximum of a particle's equilibrium saturation ratio, $S_{crit}$, as expressed by the following equation:

$$S = a_w \exp\left( \frac{4\sigma M_w}{RT\rho_w D_p} \right) \times 100\,\%. \tag{1}$$

Here, $\sigma$ denotes the effective surface tension of the particle at the air–liquid interface. $M_w$ and $\rho_w$ are the molar mass and density of water, respectively. $R$ is the gas constant, $T$ is the temperature, and $D_p$ is the diameter of the particle. The so-called solute or Raoult effect is captured by the water activity, $a_w$, whereas the exponential factor captures the Kelvin effect. These two effects are often considered to be in competition with one another since the Raoult effect decreases the supersaturation necessary for an aerosol (or cloud condensation nucleus, CCN) to reach activation into a cloud droplet while the Kelvin effect





increases the critical supersaturation. A common reason for considering this a competition is the following: as surface-active species are depleted from a droplet's interior "bulk" to populate the surface during hygroscopic droplet growth, their lowered

concentration in the bulk will typically raise the value of $a_w$ while simultaneously decreasing the value of $\sigma$ by lowering the effective surface tension. However, we note that there is not necessarily a competition. In particular, in multicomponent organic–inorganic particles with substantial nonideal mixing, such as in cases with LLPS, the bulk–surface partitioning of low-polarity surfactants can lead to both a relative lowering of surface tension as well as of $a_w$ (e.g. Ovadnevaite et al., 2017).

In an effort to simplify calculations of aerosol water uptake and CCN activation, Petters and Kreidenweis (2007) developed a

single-parameter model of hygroscopic growth, commonly known as $\kappa$-Köhler theory, wherein $a_w$ can be related to the current volume of water in the particle $V_w$ and the starting dry particle (solute) volume, $V_{dry}$, through the following equation:

$$\frac{1}{a_w} = 1 + \kappa \frac{V_{dry}}{V_w}.$$  (2)

The value of $\kappa$ for a multicomponent solution can be found from a volume-fraction-based linear weighting of the $\kappa$ values of individual components. The combination of Eqs. (1) and (2), along with an assumption about $sigma$, allows for the calculation

of critical supersaturations of aerosol particles given only their composition and dry sizes (Petters and Kreidenweis, 2007); however, the utility of $\kappa$-Köhler theory is thought to be more limited for ultrafine particles (Topping et al., 2016).

While $\kappa$-Köhler theory may provide reasonable predictions for many systems, including particles in the accumulation mode or of even larger sizes that are composed of highly soluble components, its neglect of bulk–surface partitioning treatments in combination with the high surface area-to-volume ratios of ultrafine particles is worth exploring in greater detail. There are

numerous methods for determining the surface composition at a gas–liquid interface, all of which rely on various assumptions, such as the location and dimensionality of said interface and the inclusion of various system-dependent fit parameters. One of the classical methods is the Gibbs 2-dimensional (2D) dividing surface (or plane) approach, which relates the change in surface tension of a solution ($d\sigma$) to the surface excess concentration ($\Gamma_i$) and change in chemical potential $d\mu_i$ of component $i$ in the bulk solution via (Gibbs, 1874):

$$d\sigma = -\sum_i \Gamma_i d\mu_i.$$  (3)

The term surface excess concentration may be misleading in this case since it is defined as the difference between the concentration of $i$ on the Gibbs dividing plane and that in the interior volume of a bulk phase adjacent to the plane. Thus negative values of $\Gamma_i$ are possible if $i$ partitions preferentially into the bulk of a solution and has consequently a lower concentration at the surface. The definition of the location of the Gibbs surface is of the utmost importance when utilizing this approach. Its

location is typically selected such that the surface excess concentration of the first species (usually the main solvent, e.g. water) is precisely 0, thus making that 2D plane location a system-specific, composition- and size-dependent parameter. Determining the location of the Gibbs dividing plane presents additional challenges since the location of the surface is defined to be located



within the interfacial region between 2 phases, but may not necessarily be found at the same radial position as that of the "outside" edge of the monolayer of molecules found at the boundary of a phase.

From the theoretical framework laid out by Gibbs, the semi-empirical Szyszkowski–Langmuir isotherm for bulk–surface partitioning was developed (Szyszkowski, 1908):

$$\sigma = \sigma_w(T) - A^{\text{SL}} \ln\left(1 + \frac{B^{\text{SL}}}{C_i^{\text{SL}}}\right). \tag{4}$$

Here, $\sigma_w$ is the surface tension of pure water at the temperature of interest, $A^{SL}$ and $B^{SL}$ are system-dependent fit parameters, and $C_i^{SL}$ is the concentration of solute $i$ in the bulk liquid of the system. While simpler as an approach and adequate in many system-specific cases (such as the interpretation of laboratory studies), the predictive power of this equation is limited due to the inclusion of two fit parameters for each chemical system.

A third treatment for modeling bulk–surface partitioning of different species was introduced by Jura and Harkins (1946). It combined semi-empirical fitted models relating surface concentrations to surface tension as follows:

$$\sigma = \sigma_w(T) - (A_0^{CF} - A_i^{CF}) m_\sigma. \tag{5}$$

This equation describes an insoluble compressed film (CF) at the interface between a liquid and a gas with a 2D equation-of-state for bulk–surface partitioning. $A_0^{CF}$ is the maximum surface adsorption possible, $A_i^{CF}$ is the current surface adsorption of $i$, and $m_\sigma$ is a term relating the change in surface tension to the change in surface concentration. This equation, when coupled with an isotherm relating bulk and surface concentrations, is capable of describing the surface tension as aerosol particles grow hygroscopically when exposed to increasing RH (Ruehl et al., 2016). In macroscopic systems, the approaches of Szyszkowski (1908) and Jura and Harkins (1946) assume that the enrichment or depletion of species at the particle surface has a negligible effect on the bulk particle composition; however, in ultrafine particles, depletion of surface active species from the bulk phase may become important. This bulk depletion effect has recently come under scrutiny (Prisle et al., 2010; Bzdek et al., 2020; Lin et al., 2020) and has already been considered previously by Sorjamaa et al. (2004), who considered binary systems of water and the surfactant sodium dodecyl sulfate (SDS). They noted that Köhler curve calculations should include the effect of bulk depletion since neglecting this effect led to unrealistically low critical supersaturation conditions. These results were supported by laboratory measurements of ternary water–SDS–sodium chloride systems taken by Prisle et al. (2008).

While many organic species present in the atmosphere are expected to be strongly surface-active, there are others classified as weak surfactants, including organosulfates (Hansen et al., 2015), certain components in mixtures of marine SOA and POA (Ovadnevaite et al., 2017), and aliphatic dicarboxylic acids (Ruehl et al., 2016). The surface enrichment of these compounds is difficult to predict, currently inaccessible experimentally (for airborne particles), and potentially showing complex interactions with particle size (Sorjamaa et al., 2004; Sorjamaa and Laaksonen, 2007; Davies et al., 2019; Ovadnevaite et al., 2017).



## 1.1 Prior treatments of bulk–surface partitioning in aerosol systems

There have been numerous methods developed for predicting the bulk–surface partitioning of aerosol chemical species, the following is a summary of recent works. Briefly, Sorjamaa et al. (2004) used an approach based on the 2D Gibbs dividing
surface theory and found that the depletion of water-soluble surfactants may have a substantial impact on Köhler curves through both surface tension depression and modification of the Raoult effect as surfactants are depleted from the bulk. They report that failing to account for both surface and bulk effects in growing aerosol particles may lead to under-predictions of the critical supersaturation, particularly at higher organic mass fractions. Prisle et al. (2008) used the semi-empirical Szyskowski equation (Szyszkowski, 1908) to model the bulk–surface partitioning of various fatty acids with increasing carbon chain length.
Importantly, they noted that using the surface tension of pure water while accounting for changes to the Raoult effect from bulk–surface partitioning, led to good agreement with experimental data and less costly calculations; however, the use of accurate values of $a_w$ that accounted for bulk–surface partitioning were critical. These assumptions were more thoroughly explored by Prisle et al. (2010), wherein they found that particles with at least 50 % of their mass composed of surfactant species required more accurate treatments of surface tension depression than considering $\sigma$ to be the same as that of pure water. In systems with
lower concentrations of surfactants it was noted that the surface tension may be similar to that of pure water at the point of CCN activation. Despite this, an evolving surface tension depression in growing aerosol particles that have not yet activated may play a role in their growth and activation behavior (Ovadnevaite et al., 2017; Davies et al., 2019). Further complicating the issue is a lack of measurements of surface tension on sub-500 nm diameter particles.

Romakkaniemi et al. (2011) included a Szyskowski–Langmuir treatment of bulk–surface partitioning to droplets containing
methylglyoxal, a semivolatile species and moderate surfactant. They found that treating the surface as a hypothetical 2D plane that mixes ideally led to over one order of magnitude increase in the total particle-phase concentration of methylglyoxal, further confirming the importance of bulk–surface exchange treatments. Beyond Szyskowski–Langmuir isotherm-based treatments of bulk–surface partitioning, other equations of state have been employed, including 2D van der Waals models, compressed film models, and LLPS-based models of bulk–surface partitioning (Ruehl and Wilson, 2014; Ruehl et al., 2016; Ovadnevaite et al.,
2017). The compressed film model of Jura and Harkins (1946); Ruehl et al. (2016) was also utilized by Forestieri et al. (2018) to examine long-chain fatty acid coatings on sodium chloride particles to mimic sea spray aerosols. It was found that different surfactant species may have large effects on CCN activation through both their impacts on $\sigma$ and the effective hygroscopic growth parameter $\kappa$ under high RH conditions. They note that compounds traditionally thought of as highly surface active, like the fatty acids in their study, may not have as large of an impact on CCN activation as others (Forestieri et al., 2018).
The treatment of bulk–surface partitioning was also studied using an AIOMFAC-based coupled liquid–liquid equilibrium and gas–particle partitioning calculation (Ovadnevaite et al., 2017; Davies et al., 2019). The effective surface tension was estimated based on the predicted LLPS phase compositions and the surface coverage by an organic-rich shell phase, constrained to be of a defined minimum thickness $\delta_{\beta,min}$. This approach found that LLPS aerosols can be strongly affected by surface tension reductions, yet weakly by changes to the Raoult effect following bulk–surface partitioning (Ovadnevaite et al., 2017;
Davies et al., 2019). The treatment of bulk–surface partitioning in the LLPS-based approach was primarily due to bulk equi-



librium LLPS, organic surface coverage, and gas–particle partitioning, which directly account for substantial nonideal mixing. However, other more detailed effects, such as a size-dependent feedback from bulk–surface partitioning on LLPS phase compositions and liquid–liquid–interfacial-phase partitioning and energy effects were not accounted for. Furthermore, this approach did not depend on any assumptions about the maximum thickness of the interfacial region, only a prescribed minimum thick-

ness (Ovadnevaite et al., 2017). Likewise, a subsequent study by Davies et al. (2019) compared a compressed-film model and three versions of AIOMFAC-based bulk–surface partitioning models. The first AIOMFAC-based approach involved using a full (unconstrained) liquid–liquid equilibrium calculation. This methodology assumed that the organic-rich phase $\beta$ would form a spherical shell around phase $\alpha$ if there was sufficient material, otherwise phase $\beta$ formed a partial spherical shell of thickness $\delta_{\beta,min}$ over phase $\alpha$. An area-weighted mean of pure component surface tensions based on the areas of each phase exposed to

the gas phase was used to determine the effective droplet surface tension. The second approach involved assuming a complete phase separation among organics and aqueous inorganic electrolytes, with only water allowed to partition between both phases and the assumption that the organic species formed a film, i.e. a partial monolayer or up to multiple molecular layers at the droplet surface. In case of insufficient organic material for forming a complete monolayer over the droplet, the effective surface tension of the droplet is computed as the surface coverage area-weighted average of the pure organic species surface tension

and the surface tension of water. These two treatments of surface tension provided a good and predictive estimate of the upper and lower bounds of the measured critical supersaturation of a CCN, given the initial dry composition of the particle.

Malila and Prisle (2018) developed a semi-empirical monolayer-based bulk–surface partitioning model based on an extension of an earlier method developed by Laaksonen and Kulmala (1991) wherein the surface tension of a droplet was related to a surface-composition-weighted average of pure-component surface tensions ($\sigma_i^\circ$):

$$\sigma(x^b, T) = \frac{\sum_i \sigma_i^\circ \mathscr{V}_i x_i^s}{\sum_i \mathscr{V}_i x_i^s}.$$
(6)

Here, $\mathscr{V}_i$ and $x_i^s$ are the molar volumes and surface mole fractions of $i$, respectively. Coupled with mass conservation, this equation must be solved iteratively from a given droplet size and overall composition. It is important to note that pseudo-binary approximations must be made for Eq. (6) for systems with three or more components and that the iterative solution must be optimized for each system tested, which somewhat limits the utility of this approach as a predictive tool.

Vepsäläinen et al. (2022) conducted a comparative study examining the differences between the 2D Gibbs adsorption model of Prisle et al. (2011), the simplified complete partitioning model of Prisle et al. (2011), the compressed film model of Ruehl et al. (2016), a partial monolayer model based on Ovadnevaite et al. (2017), the monolayer model of Malila and Prisle (2018), and a simple bulk-composition-based model that did not allow for bulk–surface partitioning. In their study, the hygroscopic growth of 50 nm particles of different, moderately surface-active dicarboxylic acid species were modeled. It was noted that

the more complex models for bulk–surface partitioning, those which allowed for the partial partitioning of species between the bulk and surface, had the best agreement with measured critical supersaturations. Despite the agreement between the more complex models with measured critical supersaturations as a function of dry particle size, they predicted varying degrees of bulk–surface partitioning for different species and, thus, different equilibrium compositions for the bulk and surface of the



droplet. A related model comparison by Vepsäläinen et al. (2023) was also extended to systems including stronger surfactant
species, such as myristic acid. In that recent work, it was also noted that the various bulk–surface partitioning treatments
from Vepsäläinen et al. (2022) were in agreement at low surfactant mass fractions; however, there was disagreement at higher
surfactant mass fractions.

Other recent work with thermodynamics-based models have attempted to predict the degree to which surfactants may cover
an aerosol particle (surface) at equilibrium, with many surfactant species expected to have a surface coverage on the order
of 60–85 %, while some surfactant remains dissolved in the particle bulk, both for compositions below and above the critical
micelle concentration (McGraw and Wang, 2021).

While simplifying assumptions about bulk–surface partitioning are reasonable for macroscopic systems, in the case of ul-
trafine particles, the competing effects of interior bulk phase depletion and surface accumulation are complex and must be
considered. Recent experimental findings have highlighted that even for larger droplets on the order of several micrometers
which contain non-ionic surfactant species similar to those found in atmospheric aerosols, varying surface tensions may be
exhibited across particle sizes (Bzdek et al., 2020). Thus, a rigorous framework for determining the equilibrium bulk–surface
partitioning of a droplet is necessary to better understand the size-dependency of bulk–surface partitioning and related surface
tension impacts in aerosol systems. Ideally, such a framework is predictive in design; i.e., it does not need to be fine-tuned
or fitted specifically for every system or size range of interest. In the following, we introduce and evaluate such a framework
by extending the AIOMFAC-based gas–particle partitioning calculations to include a distinct surface phase present between
aerosol particles and the surrounding gas phase; see Fig. 1.

## 2   Theory and Methods

In his seminal works, Gibbs (1874) defined the following equation capable of expressing the so-called "free" energy of a
thermodynamic system ($G$) with negligible energetic contributions from surfaces and interfaces as:

$$G = U - TS - PV. \tag{7}$$

Here, $U$ is the internal energy of the system, $T$ is temperature, $S$ is entropy, $P$ is pressure, and $V$ is the volume of the system.
Taking the derivative of Eq. (7) and then integrating with respect to the number of moles of $i$ in the system ($n_i$) at constant $T$
and $P$ gives the following equation for a system with $k$ components:

$$G = \sum_k \mu_k n_k. \tag{8}$$

Using the information given in Eq. (8), the equilibrium composition of a closed multiphase system can be found when its
energy is at its global minimum. However, in order to accurately predict the chemical potential of each component in a system,
physicochemical mixing models must be employed to account for non-ideal mixing effects on the chemical potential of $i$ in a



liquid-state bulk phase $b$ as given by:

$$\mu_i^b = \mu_i^{\circ,b} + RT\ln(a_i^b), \tag{9}$$

where, $\mu_i^{\circ,b}$ is the standard chemical potential, $R$ is the gas constant, and $a_i^b$ is the (chemical) activity of $i$. The activity of a species is a unitless value which represents the "effective concentration" of component $i$ on a chosen composition scale (e.g. mole fraction or molality) as determined by non-ideal mixing in a given phase of a system. Under equilibrium conditions, the chemical potentials of a component must be equivalent across all phases (e.g., Zuend et al., 2010). A useful choice, often employed when calculating $\mu_i$ for multiple phases of a system, is that $\mu_i^\circ$ is defined to be identical for all phases. With that

definition, also the activities of a species must be equal in all coexisting phases for a system at equilibrium. In order to accurately predict the various $a_i^b$ over a wide range of solution compositions, complex mixing models, such as the AIOMFAC model, must be employed to account for molecular interactions, shape and size differences.

The AIOMFAC model is a state-of-the-art thermodynamic mixing model capable of accurately predicting the chemical activity of species in atmospheric aerosol systems in a thermodynamically rigorous and consistent manner. The model has been

updated numerous times to include many atmospherically relevant inorganic cations, anions, and organic functional groups, so that this model can represent a large number of mixed organic–inorganic aerosol systems, including systems with tens to thousands of components (Zuend et al., 2008, 2011; Yin et al., 2022).

If two or more distinct phases (phases are here indexed by superscript $\phi$) are present in a system at constant temperature and pressure, Eq. (8) may need to be expanded to account for the energy associated with the boundaries between the phases.

This can be achieved via the addition of a term accounting for the interfacial energy per unit area ($\sigma$) scaled by the area of the interface ($A$) (e.g., Aston and Herrington, 1991):

$$G = \sum_\phi \sum_j \mu_j^\phi N_j^\phi + \sigma A. \tag{10}$$

The interfacial Gibbs energy contribution can be important for microscopic systems, in which the contribution gains in magnitude relative to the collective energy content of the bulk volume of the phases. This interfacial Gibbs energy is therefore of

interest in the context of small aerosol particles and cloud droplets. In the case of macroscopic systems, the interfacial energy term can usually be neglected due to its comparably tiny contribution.

Gibbs defined interfacial energy (or tension) as the excess energy attributed to the 2-dimensional boundary present between two phases. However, the choice to describe the boundary between two phases as a 2D surface and deciding its exact location presented several problems, because real systems may exhibit concentration gradients near an interface. To account for this,

Guggenheim introduced the concept of a 3-dimensional (3D) interfacial phase, which includes the boundary between two distinct phases and the concentration gradients of species therein (Guggenheim, 1940). While often on the order of a molecular monolayer or bilayer in depth, such an interfacial compartment can be treated as a distinct phase and must abide by the thermodynamic equilibrium conditions. Given the above-mentioned challenges when using Gibbs' 2D surface treatment and in





order to more easily connect geometrically confined interfacial and bulk phases in a mass-conserving finite-volume system, the

approach of Guggenheim's 3D interface is adopted in our framework. Thus, for application to fine and ultrafine particles, we introduce an interface with a small but finite thickness ($\delta$). We will focus in the following derivations on particles of spherical shape (droplets).

Following the approach of Cai and Griffin (2005), species adsorbed onto or absorbed into the surface of an aerosol particle contribute to the total volume of the particle. Thus, the interface only extends from the limit of the particle's radius inward and

the total number of condensed-phase molecules of $i$ ($n_i^{tot}$) is defined as the sum of the number of molecules of $i$ in the bulk ($n_i^b$) and in the surface ($n_i^s$):

$$n_i^s + n_i^b = n_i^{tot}. \tag{11}$$

If the surface of an aerosol is chemically distinct from the underlying bulk phase, then a particle with one condensed (here liquid) bulk phase, a distinct surface phase, and the surrounding gas phase, will satisfy the criterion of equivalency of chemical

potentials at equilibrium, as mentioned previously. As described by Aston and Herrington (1991), the chemical potential of a surface can be defined by differentiating Eq. (10) for a surface phase with respect to the molar amount of component $i$ in the surface $n_i^s$:

$$\left(\frac{\partial G^s}{\partial n_i^s}\right)_{T,P,n_j^s,A} = \left(\frac{\partial G^s}{\partial n_i^s}\right)_{T,P,n_j^s,\sigma} - \left(\frac{\partial G^s}{\partial A}\right)_{T,P,n_i^s} \times \left(\frac{\partial A}{\partial n_i^s}\right)_{T,P,n_j^s,\sigma}. \tag{12}$$

This leads to the following expression for the chemical potential of a surface phase:

$$\mu_i^s = \xi_i^s - \sigma \mathscr{A}_i, \tag{13}$$

where $\xi_i^s = \left(\frac{\partial G^s}{\partial n_i^s}\right)_{T,P,n_j^s,\sigma}$, $\sigma$ is the surface energy per unit surface area of the solution (i.e. the interfacial energy per unit area of the gas–liquid interface), and $\mathscr{A}_i = \left(\frac{\partial A}{\partial n_i^s}\right)_{T,P,n_j^s,\sigma}$ is the partial molar area of $i$. $\mathscr{A}_i$ can be found for a finite-depth Guggenheim surface phase of a spherical particle as follows:

$$\left(\frac{\partial A}{\partial n_i^s}\right)_{T,P,n_i^s,\delta} = \frac{\left(\frac{\partial A}{\partial r}\right)_{T,P,n_i^s,\delta}}{\frac{\partial n_i^s}{\partial r}}. \tag{14}$$

Here, the change in surface area of a sphere with respect to its radius is:

$$\left(\frac{\partial A}{\partial r}\right)_{T,P,n_i^s,\delta} = 8\pi r \tag{15}$$





and the volume of the finite-depth surface phase on a spherical droplet is:

$$v^s = \frac{4}{3}\pi(r^3 - (r-\delta)^3).$$ (16)

Differentiating Eq. (16) with respect to the particle radius assuming a constant surface depth leads to

$$\left(\frac{\partial v^s}{\partial r}\right)_\delta = 4\pi(2\delta r - \delta^2).$$ (17)

Equation (17) can be converted into an expression for $\frac{\partial n_i^s}{\partial r}$ with the inclusion of the molar volume of $i$ in the surface ($\mathscr{V}_i^s$). This value is assumed to be the same as the bulk molar volume of pure $i$, which can be calculated by dividing the molar mass ($M_i$) by the liquid-state density ($\rho_i$). This yields

$$\frac{\partial n_i^s}{\partial r} = \frac{4\pi(2\delta r - \delta^2)}{\mathscr{V}_i^s}.$$ (18)

Combining Eqs. (15) and (18) along with the assumption that the molar volumes are additive in order to determine $n_i^s$ leads to Eq. (19) for the partial molar area $\mathscr{A}_i$:

$$\mathscr{A}_i = \left(\frac{\partial A}{\partial n_i^s}\right)_{T,P,n_j^s,\sigma} = \mathscr{V}_i^s \frac{2r}{2\delta r - \delta^2}.$$ (19)

In the extreme case of the depth of the interface approaching the value of a droplet's radius, $\lim_{\delta \to r}$ of Eq. (19) reduces to $\frac{2}{r}\mathscr{V}_i^s$. In the opposite limiting case of a macroscopic system, in which $\delta \ll r$ and $\delta^2 \approx 0$, the $\lim_{r \to \infty}$ of Eq. 19 is $\frac{1}{\delta} \times \mathscr{V}_i^s$.

Integrating partial molar areas by Euler's theorem leads to

$$A = \sum_j n_j^s \mathscr{A}_j$$ (20)

for a multicomponent system with $j = 1, \ldots, k$ different species present. Note that for cases in which the surface phase extends beyond monolayer thickness, Eq. (20) indicates that the $\mathscr{A}_i$ values will not necessarily correspond to the area per molecule located directly at the 2D gas–liquid surface.

Returning to Eq. (13), the intrinsic chemical potential, $\xi_i^s$, of components in the surface phase can be formulated analogously to that of a liquid phase (Aston and Herrington, 1991):

$$\xi_i^s = \xi_i^{\circ,s} + RT\ln(a_i^s).$$ (21)





Here, $\xi_i^{\circ,s}$ is the intrinsic standard chemical potential and $a_i^s$ is the activity of $i$ in the surface phase. In the case of pure component $i$, Eq. (13) yields:

$$\mu_i^{\circ,s} = \xi_i^{\circ,s} - \sigma_i^\circ \mathscr{A}_i^\circ \tag{22}$$

where $\sigma_i^\circ$ is the surface energy per unit area of pure component $i$. For a more general multicomponent surface phase, Eq. (13) becomes:

$$\mu_i^s = \mu_i^{\circ,s} + RT\ln(a_i^s) + \sigma_i^\circ \mathscr{A}_i^\circ - \sigma \mathscr{A}_i. \tag{23}$$

Assuming that $\mathscr{A}_i^\circ \approx \mathscr{A}_i$ under all conditions, defining $\mu_i^{\circ,s} = \mu_i^{\circ,b}$, and further requiring that at equilibrium the chemical
potentials of species across phases are equal, the effective surface tension of a multicomponent solution can be isolated to form the following equation:

$$\sigma_i = \sigma_i^\circ + \frac{RT}{\mathscr{A}_i} \ln\left(\frac{a_i^s}{a_i^b}\right). \tag{24}$$

At equilibrium all $\sigma_i$ values must be equivalent to a common value, the effective solution surface tension ($\sigma$). In this case, Eq. (24) can be recognized as the Butler equation (Butler and Kendall, 1932; Sprow and Prausnitz, 1966). Employing Eq. (24)
for each component in a system, along with Eq. (11), leads to a system of equations for the equilibrium composition of a 3D Guggenheim interface. These equations can be solved iteratively in a nested manner in combination with the AIOMFAC-based gas–particle partitioning and liquid–liquid phase separation algorithms laid out in Zuend et al. (2010) and Zuend and Seinfeld (2013).

## 2.1    Calculation of Surface Composition

In order to determine the composition of the surface phase in practice, a system of equations which allows maximum freedom of variable ranges, yet simultaneously satisfies the volume balance constraints of the 3D interface, is required. Furthermore those equations must not modify variables that should be held constant during the calculation of partial derivatives. For this purpose, we have developed a conforming implementation within the extended AIOMFAC equilibrium model. We begin by introducing a term representing the fractional amount of component $i$ relative to the total amount of $i$ in the particle phase:

$$\varepsilon_i = \frac{n_i^s}{n_i^{tot}}, \tag{25}$$

where $n_i^s$ denotes the molar amount in the surface phase and $n_i^{tot}$ is the total molar amount of $i$ available for partitioning (a constant during partitioning calculations). From the values of $r_p$ and $\delta$, the total volume of the surface phase is calculated according to Eq. (16). Likewise $n_i^s$, $\rho_i$, and $M_i$ can be used to determine the volume of $i$ in the surface phase. The fraction of





the total surface volume occupied by $i$ is given by

$$f_i = \frac{v_i^s}{v^s} = \frac{v_i^s}{\sum_j v_j^s}. \tag{26}$$

We now define a new variable, $\zeta_i$, expressing the fraction of $i$ in the surface relative to its assigned volume range such that

$$\zeta_i v_i^{rg} + v_i^{min} = v_i^{s\star}, \tag{27}$$

where $v_i^{rg} = v_i^{max} - v_i^{min}$ and $v_i^{max}$ and $v_i^{min}$ are the respective maximum and minimum possible surface volume contributions. $v_i^{s\star}$ is the unnormalized surface volume contribution of $i$. The values of $\zeta_i$ can vary in the range from 0.0 to 1.0; however, there can be non-zero $v_i^{min}$ values for some components (e.g. for water at very high RH) in order to achieve volume closure between the targeted geometric surface shell volume and the (unnormalized) volume as calculated by summing up the surface quantities, such that $v_i^{min} \geq 0$ and $v_i^{max} \geq v_i^{min}$. For a given particle composition, $r_p$, and $\delta$, both $v_i^{min}$ and $v_i^{max}$ are species-specific constants. As such, $v_i^{min}$ can be determined depending on whether all other components $j \neq i$ can occupy all of the surface volume if they are at their maximum abundance in the surface. If this is the case, then $v_i^{min} = 0$. Otherwise, $v_i^{min} > 0$. This leads to

$$v_i^{min} = \max \left[ v^s - \sum_{j, j \neq i} v_j^{max}, \; 0 \right]. \tag{28}$$

Here, $v_i^{max}$ is the maximum possible volume in the surface phase, less than or equal to $v^s$, such that

$$v_i^{max} = \min \left[ n_i^{tot} \times \mathcal{V}_i, \; v^s \right]. \tag{29}$$

Returning to $\zeta_i$, Eq. (27) can be rearranged as

$$\zeta_i = \frac{v_i^{s\star} - v_i^{min}}{v_i^{max} - v_i^{min}}. \tag{30}$$

For a given value of $\zeta_i$, $v_i^{s\star}$ can be computed and then normalized to find $f_i$ via

$$f_i = \frac{v_i^{s\star}}{\sum_j v_j^{s\star}} \tag{31}$$

and a value for $v_i^s$ can be determined via

$$v_i^s = f_i \times v^s. \tag{32}$$





We note that such calculated $f_i$ and $v_i^s$ values may violate the condition that $v_i^s \leq v_i^{max}$. This issue is remedied by introducing

the relative deviation of actual to targeted surface phase volume an additional equation (constraint) to be solved alongside with

the equations describing the partitioning of $k-1$ components. From $v_i^s$, variables $n_i^s$ and $\varepsilon_i$ can be computed via $n_i^s = \frac{v_i^s}{\mathscr{V}_i}$ and

then applied in Eq. (25) to obtain $\varepsilon_i$.

## 2.2    Initial guess generation

An initial guess for the surface composition of a given particle can be derived from a first calculation for a non-partitioning

(superscript $np$) case. In this trivial case, the relative compositions of the surface and bulk phases are set to be identical, such

that $x_i^s = x_i^b$ and $f_i^s = f_i^b$. From this case, an estimation of $\zeta_i^{np}$ can be computed given that:

$$f_i^{np} = \frac{n_i^{tot}\mathscr{V}_i}{\sum_j n_j^{tot}\mathscr{V}_j} \tag{33}$$

and

$$v_i^{s\star,np} = \min\left(f_i^{np}\, v^s, \; v_i^{max}\right), \tag{34}$$

to yield

$$\zeta_i^{np} = \frac{v_i^{s\star,np} - v_i^{min}}{v_i^{max} - v_i^{min}}. \tag{35}$$

Given the definition that $a_i = x_i\gamma_i$ for neutral components (and $a_{\pm,i} = \frac{m_{\pm,i}}{m^\circ}\gamma_{\pm,i}$ for electrolytes), Eq. (24) can be rearranged

to the following form:

$$\ln\left(\frac{x_i^s}{x_i^b}\right) = (\sigma - \sigma_i^\circ)\frac{\mathscr{A}_i}{RT} - \ln\left(\frac{\gamma_i^s}{\gamma_i^b}\right). \tag{36}$$

As long as the same nonideal mixing model is used for bulk and surface activity coefficients (in this case, AIOMFAC), the

activity coefficient ratio on the right-hand side of Eq. (36) is equal to 1 (only in this non-partitioning case). For a spherical

particle of known surface volume $v^s$ as well as $f_i^s$, and $\mathscr{V}_i$ values, the non-partitioning assumption enables the calculation of

the molar phase amounts and the surface-to-bulk molar ratios, $\frac{n_i^s}{n_i^b}$. Using a composition-weighted mean of the pure-component

surface tensions for $\sigma$ allows for the evaluation of Eq. (36), which yields $\frac{x_i^s}{x_i^b}$. For neutral components (mole fraction scale), the

obtained mole fraction ratios can then be converted into a (new) guess for the set of $\varepsilon_i$ values as follows:

$$\varepsilon_i^{guess} = \frac{n_i^s/n_i^b}{1 + (n_i^s/n_i^b)} = \frac{x_i^s}{x_i^b}\frac{\sum_j n_j^s}{\sum_j n_j^b}. \tag{37}$$





For electrolyte components with activities defined on molality scale, the corresponding bulk–surface partitioning guess is generated via scaling by the surface-to-bulk phase mass ratio:

$$\varepsilon_i^{guess} = \frac{x_i^s}{x_i^b} \frac{\sum_l n_l^s M_l}{\sum_j n_l^b M_l}, \tag{38}$$

where the summation index $l$ covers non-electrolyte components (i.e. solvents) only. Using the determined set of $\varepsilon_i^{guess}$ values, Eq. (24) can be evaluated to obtain updated activity coefficient ratios, $\frac{\gamma_i^s}{\gamma_i^b}$, as well as an updated weighted-mean estimate of the surface tension, which can then once more be evaluated with Eq. (36) and processed to obtain an updated $\varepsilon^{guess}$ vector. If desired, one can expand on this approach by using the determined activity coefficient ratios (in this case kept fixed) together with a set of distinct guesses for potential equilibrium $\sigma$ values in Eq. (36), yielding a set of initial guesses for the $\varepsilon_i$ values. Systematically generating more than one initial guess is useful when the subsequent numerical solution of the system of nonlinear equations, given by Eqs. (24) or (40), is unsuccessful in case of the first initial guess evaluated – or to further explore whether more than one solution may exist. In our modern Fortran implementation, the system of equations is solved by a modified, bound-constrained version of Powell's hybrid method (Moré et al., 1980, 1984). Our extensive numerical testing suggests that this approach results in a fast and robust method for finding the equilibrium bulk–surface partitioning state for a given overall particle composition, radius and interfacial thickness.

## 2.3 Model Assumptions

A key piece of information that is necessary to solve Eq. (24) is the liquid-state pure-component surface tension, $\sigma_i^\circ$, at given temperature. In this work, $\sigma_i^\circ$ values of organic components were taken from published data (Hyvärinen et al., 2006; Riipinen et al., 2007; Booth et al., 2009) or, in the case of glutaric acid, extrapolated from high concentration data of a binary aqueous solutions (Booth et al., 2009). For inorganic electrolyte components, $\sigma_i^\circ$ was calculated using the approach for estimating pure molten salt surface tensions as described in Dutcher et al. (2010). Organic compounds with poorly constrained $\sigma_i^\circ$ values were assumed to have values of $35\ \mathrm{mJ\,m^{-2}}$. In addition, there are physical constraints applied to the surface phase thickness $\delta$ in this study. The lower limit of $\delta$ was selected to be $0.15\ \mathrm{nm}$, or the approximate length of a single alkane $\mathrm{C-C}$ bond. The upper bound for $\delta$ was selected to be $1.0\ \mathrm{nm}$ for all systems tested, which is the approximate length of 3 water molecule diameters. An alternative assumption was also tested: a treatment where $\delta$ is a function of particle or surface phase composition; however, there is limited information on how exactly $\delta$ should change as a function of composition. Therefore, in this test case it was assumed that a simple weighted average of molecular lengths based on the surface mole fraction in the particle phase could be used. These molecular lengths were computed based on $\mathscr{V}$ for each species and the assumption that the molecular length scale can be approximated by the side length of a volume-equivalent cube. Following the calculation of the surface composition, $\delta$ was updated and the surface composition was recalculated. This process was repeated iteratively until convergence (within a set tolerance) to a stable $\delta$ value. It is also important to note that Eqs. (24) and (40) cannot be solved directly for a species which is completely insoluble in either the surface or the bulk phase. As such, the relative surface tension deviations, to be solved for





from these equations, are scaled by a smooth ($\sim$ rectangular step) weighting factor expressed by the following function:

$$
\quad \Delta\sigma_i = \frac{\sigma_i - \bar{\sigma}}{|\bar{\sigma}| + \tau_\sigma} \times \frac{\varepsilon_i(1-\varepsilon_i)}{\varepsilon_i(1-\varepsilon_i) + 100\sqrt{\epsilon}\cdot\exp\left[-\sqrt{\varepsilon_i(1-\varepsilon_i)}\right]}. \tag{39}
$$

Here, $\epsilon$ is the floating point machine precision employed, $\bar{\sigma}$ the weighted mean surface tension and $\tau_\sigma$ a tolerance value, typically set to $0.1\,\mathrm{J\,m^{-2}}$. The weighting factor (expression after $\times$) on the right-hand side of Eq. (39) evaluates to near 1.0 in most cases and smoothly transitions to substantially smaller values only as $i$ becomes very close to insoluble in either the surface phase or bulk phase, in which case $\varepsilon_i$ approaches either 1 or 0. Therefore, in cases of extremely limited solubility of $i$ in
the surface or bulk, the contribution of $i$ to the system of equations used to solve for bulk–surface equilibrium is diminished (a desired property, since the numerical uncertainty grows near the limits of solubility and numerical precision limitations become substantial). We note that the weighting factor is computed prior to (but not during) numerically solving the system of equations and is only updated if deemed necessary afterwards, such as when the solver was unsuccessful for set numerical tolerances and the equations solving needed to be repeated. Furthermore, a closely related weighting factor, which is normalized by the sum
of weightings such that the resulting fractional weights sum to 1.0, is used in the calculation of the weighted mean $\bar{\sigma}$ value during the process of iteratively solving the system of equations (i.e. solving simultaneously solving Eq. (39) via Eq. (24) for all components).

An additional assumption made in this study concerns the computation of activity coefficients. It is assumed that there are no modifications to the calculation (by AIOMFAC) of activity coefficients in the surface phase compared to the bulk phase. Lane
(1983) introduced a common exponential correction factor, $t$, that is applied to the calculated activity coefficients of a surface phase, such that $\gamma_i^s = \left(\gamma_i^{s,calc}\right)^t$. The introduction of such an exponent is motivated by the idea that activity coefficients in a surface, affected by some limitations in the molecular packing options, may deviate slightly from those calculated for a bulk phase of identical composition and temperature. However, estimated values of $t$ are system-dependent, yet often close to unity Lane (1983). Because the inclusion of additional semi-empirical terms limits the flexibility of the targeted predictive capability
of the model developed in this work, it is assumed that $t = 1$ for all systems.

Returning to Eq. (24), if an alternate assumption is made about the partial molar areas, specifically that $\mathscr{A}_i^\circ \neq \mathscr{A}_i$, then the expressions previously leading to Eq. (24) result in:

$$
\sigma_i = \sigma_i^\circ \frac{\mathscr{A}_i^\circ}{\mathscr{A}_i} + \frac{1}{\mathscr{A}_i}RT\ln\left(\frac{a_i^s}{a_i^b}\right). \tag{40}
$$

and $\mathscr{A}_i^\circ$ can be found for a pure droplet of $i$ analogous to Eq. (19), with the only modification being that the interfacial thickness
used is that of the pure component, i.e. set $\delta = \delta_i^\circ$. If it is assumed that only a monolayer of molecules form the surface of a pure-component droplet then $\delta_i^\circ$ can be estimated based on the molecular size of $i$. With this information, Eq. (40) can be employed with Eq. (11) and Eq. (39) in the same manner as Eq. (24) to form a system of equations for solving numerically the equilibrium bulk–surface partitioning problem.



## 3 Results and discussion

### 3.1 Comparison of measured and predicted surface tension

In order to determine the validity of the described predictive model, comparisons were made to measurements for a selection of atmospherically relevant binary systems. Figure 2A shows the predicted surface tension (utilising AIOMFAC with Eq. 24) as a function of the total particle-phase concentration of glutaric acid in a binary water–glutaric acid droplet that was allowed to grow hygroscopically from a starting dry diameter of 5 μm to the point of cloud droplet activation, corresponding to a diameter of approximately 10 μm. This scenario allowed us to compare predicted surface tensions to the bulk tensiometry and optical tweezers measurements taken by Bzdek et al. (2016). The three curves shown in Fig. 2A correspond to three different values of $\delta$: 0.1 nm, which is the approximate length of a single carbon–carbon bond; 0.3 nm, which is the approximate length scale of a single water molecule (the same value was also used as a minimum thickness in models by Davies et al. (2019) and Ovadnevaite et al. (2017); and 1.0 nm, which corresponds to the approximate size of a single glutaric acid molecule along its longest axis. It is shown that assuming a thinner interfacial thickness value leads to a surface tension curve which is highly sensitive to the overall concentration of glutaric acid in the droplet. Analogously, an interfacial thickness that is larger will require greater changes in the total particle-phase concentration of an organic species in order to observe a similar decrease in surface tension. In addition to the binary water–glutaric acid system, Bzdek et al. (2016) also analyzed a binary water–sodium chloride system of the same size using the bulk tensiometry and optical tweezers approaches. Shown along with these data in Fig. 2B are bulk measurements by Ozdemir et al. (2009). While the surface tension increases with salt concentration in this case, a similar behavior can be seen in the surface tension vs. concentration curves for the inorganic electrolyte system, wherein the curve is sensitive to the selection of $\delta$. However, for the system shown in Fig. 2B, the sensitivity of the modeled surface tension to the value of $\delta$ is opposite of that for surface-active organic species. This is likely due to the fact that electrolytes preferentially partition into the bulk phase; therefore, a thinner interface will contain a higher mole fraction of water (at a specific wet diameter) and requires greater concentrations of electrolytes to increase a droplet's surface tension. Additional sensitivity comparisons were performed to determine the effect of modifying $\sigma_i^\circ$ by $\pm 10\%$. It was found that glutaric acid was more sensitive to modifications in the value of $\sigma_{glutaric}^\circ$ than NaCl, which was very weakly sensitive to increases in $\sigma_{NaCl}^\circ$ and somewhat sensitive to reductions in $\sigma_{NaCl}^\circ$. Overall, using $\delta = 0.3$ nm leads to surface tension predictions in better agreement with the measurements shown for large droplets or macroscopic solutions. This suggests that a molecular monolayer assumption for representing the surface phase is a relatively good model, at least for the systems shown in Figs. 2A and 2B.

Figures 2C and 2D show the effects of exploring different assumptions of Eq. (24) along with the same measurements from Figs. 2A and 2B. The modified Butler equation, which assumes that $\mathscr{A}_i^\circ \neq \mathscr{A}_i$ (Eq. 40 in this work) consistently overestimates the surface tension of the water–glutaric-acid-system and only agrees with the water–NaCl system at very low concentrations. Assuming that $\delta$ is surface composition-dependent also leads to poor agreement with experimental data at high concentrations for the water–glutaric-acid system and better agreement at low concentrations. For the water–NaCl system, it can be seen that the opposite is true; the composition-dependent $\delta$ leads to good agreement at higher concentrations than lower concentrations.



The combination of Eq. (40) and a surface composition-dependent $\delta$ leads to poor agreement across all concentrations shown for the water–glutaric-acid system and only gives good agreement for the water–NaCl system at low concentrations.

It is important to note that the sizes of the droplets analyzed in Bzdek et al. (2016) are large relative to the more numerous but substantially smaller atmospheric aerosol particles of importance in cloud formation. However, there is a paucity of surface tension data available for droplets in the sub-500 nm size range, since it is extremely difficult to measure the surface tension of atmospherically relevant aerosol particles that are freely suspended while at sizes near or below the wavelengths of visible light.

### 3.2 Bulk phase depletion

While the measured particles associated with the data from Fig. 2 are large, the effects of bulk–phase depletion cannot be neglected entirely. Figures 3A, 3B, and 3C show surface tension as a function of particle size and $\delta$ for binary water–adipic acid systems of particles with dry diameters of 1.0 nm, 10 nm, 100 nm, 1 μm, 10 μm, 100 μm, 1 mm, and 1 cm (ranging over 8 orders of magnitude). It is shown that even highly dilute particles with dry diameters below 100 μm have different surface tensions at the same total mole fraction of adipic acid in the condensed phase. If the value of $\delta$ decreases, the surface tension curves for larger dry diameters converge, and bulk phase depletion is only noticeable for the smallest particles. If $\delta$ is increased to be more similar to the molecular length scale of the solute species ($\sim 0.45$ nm), there is better agreement between modeled and measured surface tension values in the larger sized droplets. Increases in $\delta$ also increase the minimum concentration of solute necessary to decrease the surface tension from that of pure water for all particle sizes; however, smaller particle sizes are more responsive to this change than larger ones. A method for determining this dependence is by taking the concentration of solute ($c_i^{\mathrm{d}\sigma}$) at which the change in surface tension with each additional molecule of solute added is the greatest; in other words, the global extrema (minimum for surfactant species and maximum for tensoionic species) of $\frac{\mathrm{d}\sigma}{\mathrm{d}X_i^{total}}$. Figure 3D shows how $X_i^{\mathrm{d}\sigma}$ varies as a function of size for the binary water–adipic acid system. Droplets with initial dry diameters on the order of 1 μm may still experience mild bulk-depletion effects depending on the selection of the interfacial thickness value, with larger $\delta$ values leading to more pronounced bulk-phase depletion effects at larger sizes. Moreover, we note that the value of $X_i^{\mathrm{d}^2\sigma}$ is strongly dependent on $\delta$ at larger sizes compared to smaller ones, with a doubling of the interfacial thickness from 0.15 nm to 0.3 nm leading to approximately 3 orders of magnitude increase in the value of $X_i^{\mathrm{d}\sigma}$ for particles with dry diameters larger than 1 μm. This same change in $\delta$ for particles with dry sizes below 100 nm leads to differences of about 2 orders of magnitude or less. Thus, we demonstrate that bulk-phase depletion may modify surface tension as a function of particle composition on particles up to the micrometer size scale.

the bulk–surface concentration ratio $\frac{x_i^{surf}}{x_i^{bulk}}$ is independent of the volume ratio and represents the physicochemical partitioning coefficient. Figure 4 shows these mole fraction ratios for each species in a ternary water–glutaric-acid–sodium-chloride system as a function of equilibrium saturation ratio ($S$). Monodisperse particles consisting of a 1:1 molar ratio of glutaric acid to sodium chloride, with dry diameter values ranging from 10 nm to 500 nm, were allowed to grow hygroscopically until the point of cloud droplet activation (while maintaining the same solute masses, i.e. no gas–particle partitioning of glutaric acid considered). In this test, a forced 1-phase calculation was performed (only allowing a single bulk liquid phase to exist), which



prevented the particles from undergoing LLPS at compositions where that would be favorable. The value of $\delta$ was held constant at 0.3 nm and Eq. (24) was employed. It can be seen that glutaric acid, while only a weakly surface-active compound compared to lower-polarity organics, is nevertheless strongly enriched in the surface of the particle across all particle sizes, especially at higher values of $S$. The predicted surface tension for each particle is also shown. The difference in $\sigma$ of particles is largest at

525 both very low and very high values of $S$ with all of the $\sigma$ values being most similar for saturation ratios between 0.675 and 0.725 . However, particles with diameters below 25 nm exhibit greater deviations from the behavior of their larger counterparts. These smallest particles both achieve a slightly lower minimum surface tension and slightly higher surface tension values under low $a_w$ conditions. This may be driven by the fact that under these conditions, more tensoionic species must be present in the surface due to limited amounts of both water and organic species. However, this effect is still quite weak even at such small

particle sizes.

If Eq. (40) is employed for bulk–surface equilibrium predictions, the $\delta_i^\circ$ values for water, glutaric acid, and sodium chloride are calculated based on their molecular sizes. This leads to modified bulk–surface partitioning behavior as shown in Figs. 5A and 5B. The assumption that $\mathscr{A}_i^\circ \neq \mathscr{A}_i$ effectively modifies the value of $\sigma_i^\circ$ by multiplying it by the ratio $\frac{\mathscr{A}_i^\circ}{\mathscr{A}_i}$. In the macroscopic case, where $r \gg \delta$, then $\lim_{r \to \infty} \frac{\mathscr{A}_i^\circ}{\mathscr{A}_i} = \frac{\delta_i^\circ}{\delta}$, showing again an important effect of the calculated or assumed value of $\delta$. In the

535 case of species for which $\delta_i^\circ < \delta$, a stronger affinity for the surface results, while for $\delta_i^\circ > \delta$, it is expected that species $i$ would have a weaker affinity for the surface phase. Notably, the feedbacks on the partitioning behavior of one species can still modify the bulk–surface partitioning of other species since modifications to the composition of the surface phase will modify the activity coefficients for all species in the phase. As such, in Fig. 5, it can be seen that the assumption that $\delta = \delta_{water}^\circ$ leads to decreased partitioning of water as both glutaric acid and sodium chloride show an increased affinity for the surface phase.

This also modifies the effective solution surface tension. In the case where $\delta = \delta_{water}^\circ$, $\sigma$ is consistently lower than the values predicted by Eq. (24). In the case where $\delta$ lies between all of the values of $\delta_i^\circ$, $\sigma$ predicted by Eq. (40) is lower than the value predicted by Eq. (24) when the solution is highly concentrated in both organic and inorganic solutes (low $a_w$) and higher than the Eq. (24) in more dilute cases (high $a_w$). These findings suggest that it is best to assume that $\mathscr{A}_i^\circ \approx \mathscr{A}_i$ for most systems.

### 3.3 Köhler curves

Numerous different test systems have been used in laboratory experiments and modeling studies to better understand the role of bulk–surface partitioning on the behavior of CCN both before and after activation. One such system that will now be considered has been studied in experiments and theory by Ruehl et al. (2016). They generated 150 nm (dry diameter) mixed suberic acid–ammonium sulfate particles corresponding to a spherical 50 nm diameter ammonium sulfate core coated with a 50 nm layer of suberic acid. Figure 6A shows the Köhler curve calculated via Eq. (1) with the assumption that $a_w$ in that

equation is determined from the bulk phase composition of the particle. Figure 6B also shows a particle with the same organic volume fraction as Figure 6A, but of a water-free diameter of 40 nm instead of 150 nm. Different interfacial thicknesses ($\delta$) are shown including the values determined by the compressed film model used in Ruehl et al. (2016). The resulting behavior of the Köhler curve is highly sensitive to the value of $\delta$; particularly as $\delta$ approaches the lower limit of physically realistic values. The shape of these Köhler curves are determined by the point at which the surface tension of the particle becomes



similar to that of pure suberic acid relative to a given saturation ratio. In systems with $\delta$ values that are smaller the surface remains is enriched in suberic acid at higher $S$ values than systems with larger $\delta$ values. This lowered surface tension at high $S$ values may lead to modifications of the shape of the Köhler curve, including the branch at sizes greater than the critical wet diameter for CCN activation. Such behavior can be seen in the blue curve of Fig. 6A . Also shown in panels A and B of Fig. 6 are AIOMFAC-based predictive treatments of bulk–surface partitioning and CCN activation for such systems, as discussed in prior work (Ovadnevaite et al., 2017; Davies et al., 2019). These prior treatments both assume a surface with adjustable depth, consideration of LLPS and, in the presence or absence of bulk LLPS at higher $a_w$ values, the calculation of an effective surface tension using a volume-fraction-weighted mixing rule of the pure-component surface tension values based on the bulk liquid phase compositions (and area fractions in case of LLPS) in contact with the droplet surface. In this case, both simulations used $\delta = 0.3$ nm and the same pure-component surface tension values as used in the more detailed approach developed in this work. The AIOMFAC-Equil. prediction leads to substantially higher critical supersaturations than observed, because this model variant ignores bulk–surface partitioning in the case of a single bulk phase present at higher RH, as in the system of Fig. 6. In contrast, the AIOMFAC-CLLPS variant with an imposed organic film assumption agrees reasonably well with the measured critical supersaturations. Indeed, if $\delta$ is assumed to be $0.3$ nm for both prior AIOMFAC-based treatments (in those cases setting the minimum thickness of the surface phase), then the calculation by the AIOMFAC-CLLPS variant with an organic film represents the measured data better than the approach laid out in this work. However, if the $\delta$ value is lowered, then the bulk–surface equilibrium approach from this study shows excellent agreement with the measured peak supersaturation and, importantly, does not rely on the same simplifying assumptions as the organic-film-based calculation. Figure 6B demonstrates the effect of particle size (dry diameter of 40 nm) and bulk–surface partitioning on CCN activation, since the deviations from classical Köhler curve behavior (using a fixed surface tension, that of pure water) are more pronounced in both the $\delta = 0.3$ nm and $\delta = 0.15$ nm cases. In the example of Fig. 6, the AIOMFAC-Equil. prediction is representative of a classical Köhler curve, since that model does not predict LLPS in the high-$a_w$ range close to the CCN activation point for this system. Clearly, in the case of ultrafine particles, a more detailed treatment of bulk–surface partitioning and associated surface tension evolution leads to notable deviations from classical behavior. Indeed a more detailed and thermodynamically rigorous treatment of bulk–surface partitioning leads to better agreement of the activation conditions in comparison to laboratory studies of water–suberic-acid–ammonium-sulfate particles than previous AIOMFAC-based treatments.

To further demonstrate the predictive power of the model developed in this work, an isoprene SOA system was considered as well. This system consists of water and 21 semivolatile isoprene photo-oxidation products, as proposed for a simplified isoprene-derived SOA representation in previous modeling work (Rastak et al., 2017; Gervasi et al., 2020). Because of the higher volatility of some of the isoprene SOA species, the effects of organic co-condensation (or more generally gas–particle partitioning) during hygroscopic growth should also be analyzed to better understand atmospheric implications of such aerosol systems. Concentrations simulated by the Master Chemical Mechanism (Jenkin et al., 2015, 1997, 2012) were used as inputs for the "co-condensation enabed" case (Rastak et al., 2017) (see table S3 in the SI). A distinct case, in which the organic composition of the particle was fixed, termed the "co-condensation disabled" case, was also used for comparison. In that case, the equilibrium composition of the particle was taken from a bulk equilibrium gas–particle partitioning calculation at $0.1\,\%$ RH





and then fixed for the organics (essentially rendering them nonvolatile), such that only water could partition between the gas and particle phases in subsequent computations. We note that for this isoprene SOA system, the values of the various $\sigma_i^\circ$ have not been measured. We therefore assumed that $\sigma_i^\circ = 35 \ \mathrm{mJ \, m^{-2}}$ for all species, which is in line with the assumptions made in Davies et al. (2019); Ovadnevaite et al. (2017) (see SI figure S7 for an analysis of the framework sensitivity to $\sigma_i^\circ$ values). Based on experiments as well as AIOMFAC LLPS equilibrium computations for bulk solutions, this system is not expected to

undergo LLPS at any RH (Rastak et al., 2017).

Figure 7A shows the Köhler curve for a particle of 25 nm in dry diameter with co-condensation enabled and disabled. Figure 7B shows the contribution of the Raoult effect for both systems shown in Fig. 7A. Likewise Fig. 7C shows the contribution of the Kelvin effect for the same system. The inclusion of co-condensation of organic species leads to substantial reductions in $S_{crit}$ for this system through modifications to both the Raoult effect and the Kelvin effect.

# 4 Theoretical and atmospheric implications

## 4.1 Theoretical implications

One source of uncertainty in the approaches to bulk–surface partitioning described in Sect. 1 is determining the effective partial molar area of a given species, $\mathscr{A}_i$, in a mixed surface phase and how that value may differ from the molar area of pure $i$, $\mathscr{A}_i^\circ$. A common assumption is that the apparent molar area can be calculated from the molar volume ($\mathscr{V}_i$) of a species as $\mathscr{A}_i = \mathscr{V}_i^{\frac{2}{3}}$ and

605 that the molar area of a species in solution is the same as its pure-component value $\mathscr{A}_i \approx \mathscr{A}_i^\circ$. Table 1 lists the partial molar areas of numerous organic species calculated using Eq. (19) for particles of three distinct diameters and $\delta = 0.15 \ \mathrm{nm}$ or $\delta = 1.0 \ \mathrm{nm}$. Those diameter choices serve to demonstrate a size dependence in this parameter when calculated via Eq. (19). Also shown are the size-independent partial molar areas when computed under the $\mathscr{A}_i = n_A^{\frac{1}{3}} \mathscr{V}_{b,i}^{\frac{2}{3}}$ assumption or with the empirical approach developed by Goldsack and White (1983):

$$\mathscr{A}_i = 1.021 \times 10^8 \times \mathscr{V}_{c,i}^{\frac{6}{15}} \mathscr{V}_{b,i}^{\frac{4}{15}}, \tag{41}$$

where $\mathscr{V}_{c,i}$ and $\mathscr{V}_{b,i}$ are the critical and bulk molar volumes. It should be noted that the scaling factor in Eq. (41) requires that the values of $\mathscr{V}_{c,i}$ and $\mathscr{V}_{b,i}$ are input in units of $\mathrm{cm^3 \, mol^{-1}}$; the equation then returns $\mathscr{A}_i$ in units of $\mathrm{cm^2 \, mol^{-1}}$. In the mathematically sound framework developed in this work, the values of $\mathscr{A}_i$ are weak functions of particle radius $r_p$; however, they are stronger functions of the value of $\delta$, with smaller $\delta$ values leading to smaller $\mathscr{A}_i$ values.

Another important assumption made regarding the treatment of $\mathscr{A}_i$, is the assumption that the density of $i$ in the surface phase ($\rho_i^s$) is equivalent to that of the pure component value of a (bulk) liquid state. Currently, all AIOMFAC-based bulk solution calculations make the assumption that the $\mathscr{V}$ and related $\rho_i$ of a species do not change as a function of solution composition and that the total volume of a phase is a linearly additive function of the individual component molar volumes times their molar abundance. Other studies have noted that $\rho_i^s$ may differ from $\rho_i^b$ and that lower values of $\rho_i^s$ may lead to better agreement

between surface tension models and experimental data (Defay et al., 1966).



It should also be noted that deviations in activity coefficients are possible when comparing surface versus bulk phases of the same molar composition. As mentioned in Sect. 2.3, the introduction of a single exponential scaling factor ($t$) for all component activity coefficients in the surface phase has been used in the past as a fit parameter in order for binary solution surface tension curves to better match experimental data without violating the Gibbs–Duhem relation. This $t$ value may be thought of as treating the surface phase nonideality as taking place at a different temperature than that of the bulk phase, since $RT \ln [(\gamma_i^s)^t] = R(tT) \ln [(\gamma_i^s)]$. Therefore, one may argue that the value of $t$ should be constrained such that the temperature change remains physically realistic. Thus, a value of $t = 25$ is physically unrealistic for a system at $298$ K which shows substantial nonideal mixing, since it would mean that the surface phase nonideality would be behaving as if it were at $7,450$ K. The inclusion of $t$ also introduces an additional fit parameter that is likely unique for each system, thus limiting the predictive power of the model introduced in this work. Use of exponent $t$ in combination with an unconstrained fit leads to better agreement with points below cloud droplet activation for some of the Köhler curves presented in Ruehl et al. (2016), such as a ternary water–succinic–acid–ammonium-sulfate, water–pimelic-acid-ammonium-sulfate, and water–glutaric-acid-ammonium-sulfate particles. However, those fitted $t$ values must be combined with rather extreme values of $\delta$ and $\sigma_{org}$ for good agreement with the experimental data at both the CCN activation point and at points below activation (see SI Figs. S4–S6 for examples). In addition, the framework laid out in this work is incapable of simultaneously matching the growth data points and critical supersaturation point reported by Ruehl et al. (2016) for water–malonic-acid-ammonium-sulfate particles (see SI Fig. S3). Further explorations of variations in the activity coefficients of species in the surface phase are warranted to better understand how these activity coefficients may differ in value from those of a bulk solution with the same molar composition. Likewise, the explicit treatment of the dissociation of organic acids under dilute aqueous conditions is not considered in regards to its role in bulk–surface partitioning in this study. Under highly dilute conditions, such as those found in activating CCN, many dicarboxylic acids may partially or fully dissociate. It is possible that consideration of such acid dissociation may lead to modifications of both the surface enrichment and bulk depletion of different species as well as enhancements of the solute effect via an increase in dissolved ionic species. Explicit treatments of organic acid dissociation and resulting interactions among various additional ions in an AIOMFAC-based model framework are thus a direction to be explored in future work.

## 4.2 Atmospheric implications

The effect of bulk–surface partitioning on Köhler curve shapes is evident for submicron-sized aerosol particles. Even the use of relatively "thick", yet reasonable, $\delta$ values ($> 0.5$ nm) still exhibit substantial suppression of the critical supersaturation and modifications to the shape of the Köhler curve for the particle size range prior to reaching the CCN activation point under growth conditions. The inclusion of the thermodynamic treatment of equilibrium bulk–surface partitioning outlined in this study leads to simulated droplets that will grow to larger diameters at lower relative humidities than classical Köhler theory would otherwise suggest. If the value of $\delta$ is lowered, Köhler curves may exhibit a second local maximum as the CCN surface tension approaches that of pure water after the point of droplet activation. This behavior suggests that particles with very thin surface phases are more likely to activate into cloud droplets (for a given dry diameter). Clouds which form from rising air parcels populated by surfactant-containing particles may exhibit substantially higher cloud droplet number concentrations than



those forming from air parcels of comparable particle size distributions, but lack in aerosol particles of lowered (yet evolving) surface tensions. Figure 8A shows the critical supersaturation for CCN activation of particles with the same condensed phase composition as those of Fig. 6, with $D_{dry}$ values ranging from 25 nm to 130 nm. Similarly, Fig. 8B shows particles that grow under the same input parameters as used for Fig. 7, with dry sizes from 25 nm to 130 nm. The colored horizontal bands shown in Fig. 8 correspond to the typical supersaturation values experienced by aerosol particles in marine (blue), clean

continental (green), background (orange), and urban polluted (brown) cloud base conditions, according to the classification by Pinsky et al. (2014). If the AIOMFAC-Equil. model is used to determine the CCN activation conditions, particles with $D_{dry}$ between approximately 60 nm and 75 nm would not be predicted to activate in continental clouds for both systems shown. For the isoprene-derived SOA system, in particular, the inclusion of both equilibrium co-condensation of SVOCs and bulk–surface partitioning leads to larger modifications in predicted $S_{crit}$, especially for $D_{dry} > 50$ nm. This can have important

implications for both the radiative forcing effects of resulting clouds and for the precipitation formation microphysics in these clouds, since lowering the critical dry diameter for typical peak supersaturation experienced at cloud base conditions may lead to substantially increased cloud droplet number concentrations (depending on the present aerosol number–size distribution) (Ovadnevaite et al., 2017). It is also important to note that the new framework introduced in this study to date only considers impacts of organic components on CCN activation, yet does not consider the role that bulk–surface partitioning may play (if

any) for ice nucleating particles in cirrus clouds or mixed-phase clouds. Likewise, the interplay of bulk–surface partitioning and co-condensation with liquid–liquid equilibria is not yet considered in a fully coupled manner. For example, in the case of phase separation, such interactions may influence particle morphology and the RH range within which LLPS occurs. Consequently, coupled size and LLPS effects may also change the interactions between aerosol particles and light.

## 5   Conclusions

The surface area to volume ratio of atmospheric aerosol particles increases substantially as particle diameter decreases in the fine and ultrafine size ranges. Any unique properties of the exterior surface of an aerosol particle must be accounted for in order to accurately model the behavior of the smallest (sub-100 nm-sized) particles. This study builds on the finite-depth Guggenheim surface phase treatment in combination with variants of the Butler equation and AIOMFAC-based vapor–liquid equilibrium computations to create a thermodynamically rigorous treatment of bulk–surface partitioning in spherical aerosol

particles with diameters as low as 10 nm. This model relies on one adjustable, loosely constrained parameter, the surface phase thickness $\delta$, and applies consistently to any number of species in multicomponent organic–inorganic aerosol systems. The approach is capable of representing experimentally measured surface tension data for atmospherically relevant systems across a range of relative humidities. The inclusion of a thermodynamically sound treatment of interfacial regions leads to modified Köhler curve predictions that are in agreement with measured data, including for cases for which simpler approaches with fixed

surface tension fail. For particles with diameters larger than $\sim 100$ nm, the simpler AIOMFAC-CLLPS model variant with an organic film assumption agrees reasonably well with the more thermodynamically sound model in terms of predicted $SS_{crit}$ values and may serve as a good approximation when computational efficiency is a key concern. For smaller particles, where



bulk phase depletion may play a larger role, larger disagreement arises between the AIOMFAC-CLLPS model with organic film and the approaches laid out in this work.

While measurements of physicochemical properties of particles, such as surface tension and chemical composition, in the size range below 1 µm and especially below 100 nm are rare or nonexistent, there have been numerous measurements made on larger particles. Models trained on data from bulk measurements and large microscopic droplets have been used to study sub-100 nm particles. Frequently, those measurements were done on droplets exhibiting a single liquid (bulk) phase and spherical shape when freely suspended, but phase separation and associated phase boundaries can affect the particle shape. Indeed, many

systems have been modeled or observed to adopt more complex, non-spherical morphologies, in some cases involving multiple liquid phases (Huang et al., 2021; Kwamena et al., 2010; Reid et al., 2011). The basic thermodynamic theory introduced in Sect. 2 is generally applicable; however, we show that when applied to finite-volume droplets, geometric considerations introduce shape and mass-balance constraints which impact the bulk–surface partitioning, particularly in submicron-sized particles. In this study, we have outlined the detailed expressions for spherical single-bulk-phase particles, which were implemented

in our AIOMFAC-based bulk–surface partitioning model. To date, this model does not account for non-spherical shape or feedback effects from energy stored in liquid–liquid interfaces. However, our model provides a basis for future extensions to account for size-dependent feedback between droplet size, liquid–liquid interfaces, non-sphericity, and size effects on LLPS onset. Furthermore, the use of Eq. (24) or Eq. (40) for gas–liquid interfaces requires accurate measurements or predictions of a reference state values of $\sigma$, usually pure-component surface tension, for many atmospherically relevant species. These data do

not exist or, in the case of inorganic electrolytes, are disputed as to what the correct value should be. This may limit the systems for which our approach can be used – or requires assumptions to be made. Therefore, this study highlights the need for, and benefit of, reliable data for pure-component surface tension. A key goal for applications of thermodynamic multiphase aerosol models in atmospheric chemistry is achieving predictive capability, unrestricted by system- and size-dependent fit parameters. The model introduced here marks a major step toward this goal. It enables us to better quantify the role of interfacial properties

in environmental systems on the nanometer and micrometer size scales. Models like those introduced in this study can serve as a bridge between the measurable particle size range and the presently experimentally inaccessible ultrafine size range of interest for cloud formation.

*Code and data availability.*    The experimental data and model outputs underlying all figures shown are provided in an electronic supplement. The bulk–surface partitioning code is available upon request from the authors.

*Author contributions.*    RS performed model development and simulations, visualized model outputs, and wrote the manuscript. AZ conceived the project, assisted with model development, co-wrote the manuscript, was responsible for supervision, and secured funding for the work.



*Competing interests.* The authors have no competing interests to declare.

*Acknowledgements.* We would like to acknowledge Judith Kleinheins, Nadia Shardt, Claudia Marcolli, and Thomas Peter for their helpful feedback and discussions in preparation of this manuscript.

*Financial support.* This project was undertaken with financial support by the government of Canada through the federal Department of Environment and Climate Change (grant no. GCXE20S049) and the Natural Sciences and Engineering Research Council of Canada (NSERC grant no. RGPIN-2021-02688). RS was also supported by a doctoral scholarship from the Fonds de recherche du Québec – Nature et technologies (FRQNT, scholarship no. 314186) and a Mitacs Globalink research award (IT27170).



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



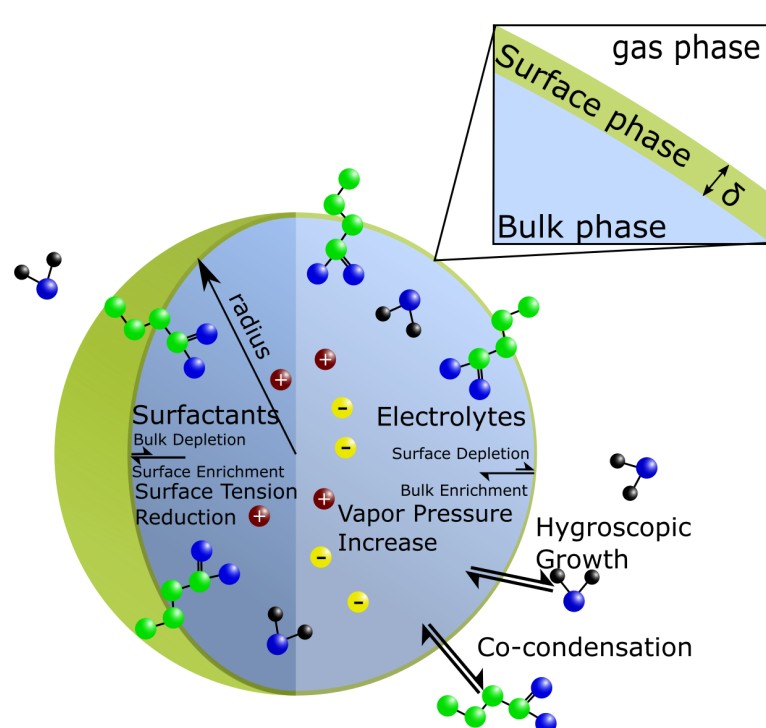

**Figure 1.** Conceptual diagram of bulk–surface partitioning in a single-bulk phase spherical aerosol particle or cloud droplet. The thickness of the finite-volume surface phase is represented by $\delta$.





**Figure 2.** Predicted and measured surface tensions for (A) a binary water–glutaric acid system and (B) a binary water–sodium chloride system as a function of solute concentration in water at 298 K. The area shaded in grey represents the predicted surface tension bounded by the proposed limits for $\delta$ of 0.1 nm and 1.0 nm. The solid black line is the predicted surface tension for $\delta = 0.3$ nm. Also shown in (C) and (D) are the same systems as (A) and (B) respectively but with the assumption that $\mathscr{A}_i \neq \mathscr{A}_i^\circ$ leading to a modified form of the Butler Equation (Eq. 40) and the testing the surface composition based approach for determining $\delta$.

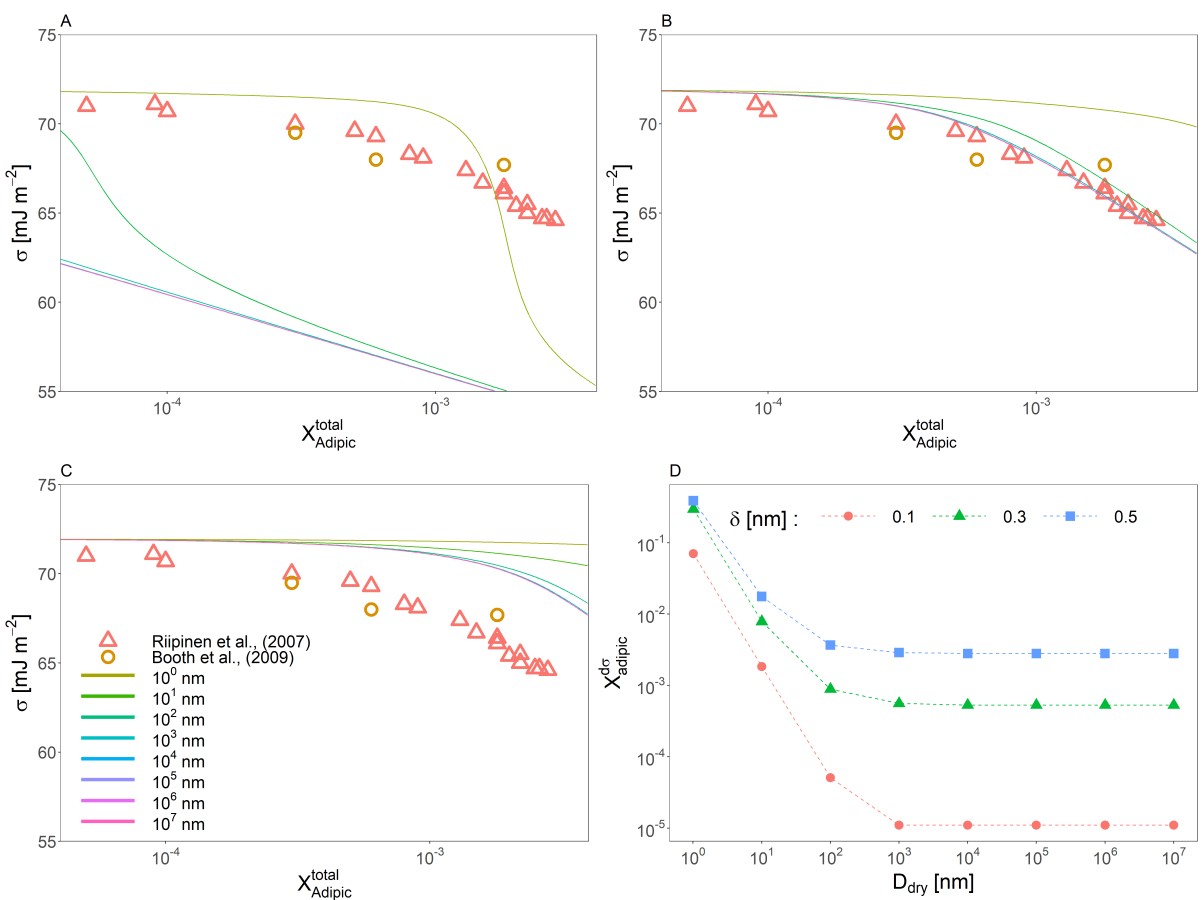

**Figure 3.** Predicted surface tensions of a binary water–adipic acid system across 8 orders of magnitude in dry particle diameter (see legend in panel C) for interfacial thickness values of (A) 0.1 nm, (B) 0.3 nm, and (C) 0.5 nm. (D) The specific concentrations of adipic acid for which the surface tension is the most sensitive to changes in the overall mole fraction of adipic acid as a function of dry diameter and $\delta$. Also shown in (A), (B), and (C) are measurements taken by Riipinen et al. (2007) and Booth et al. (2009).

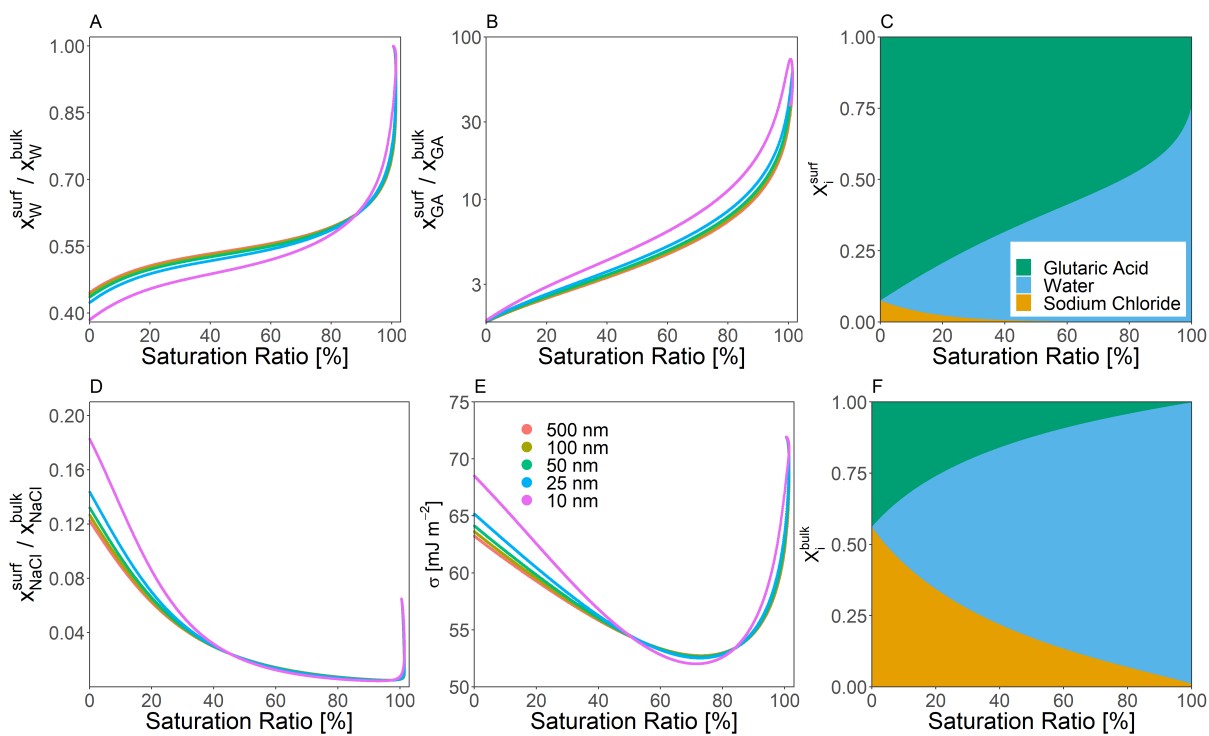

**Figure 4.** Predicted bulk–surface partitioning coefficient ($\frac{x_i^{\mathrm{surf}}}{x_i^{\mathrm{bulk}}}$) of (A) water, (B) glutaric acid, and (D) sodium chloride present in a forced single-bulk-phase particle at $T = 298$ K. A molar dry solute ratio of 1:1 was used in all cases. (E) Predicted effective surface tension for several particle dry diameters as indicated by color. Right column (composition bar graphs): shown are the mole fractions of each species in the surface and the bulk phase ($\alpha$) for a particle of 25 nm dry diameter.



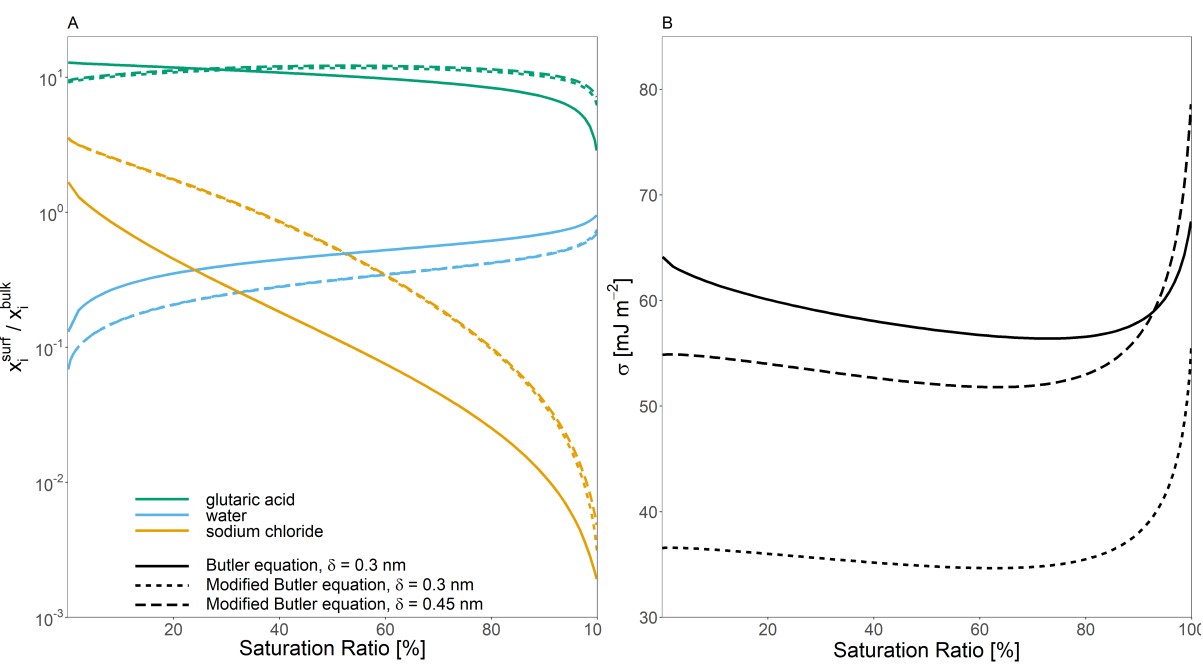

**Figure 5.** The effect of implementing a modified form of the Butler equation (Eq. 40) with the assumption that $\mathscr{A}_i^\circ \neq \mathscr{A}_i$ for a water–glutaric acid–sodium chloride system with a 1:1 water-free molar ratio of glutaric acid to sodium chloride on (A) the bulk–surface partitioning coefficient, $\frac{x_i^{\text{surf}}}{x_i^{\text{bulk}}}$, and (B) predicted surface tension, $\sigma$. All calculations were performed at a temperature of 298 K





**Figure 6.** Köhler curves for ternary water-suberic acid-ammonium sulfate particles at 298 K at an organic volume fraction of 0.88 and with a water-free diameter of (A) 150 nm (B) 40 nm. Also shown in (A) are the measured $SS_{crit}$ and the critical wet diameter measured by Ruehl et al. (2016). (C) and (D) show the calculated Köhler curves if the assumption is made that $\mathscr{A}_i^\circ \neq \mathscr{A}_i$ leading to the modified Butler equation (Eq. (40)), as well as the effect of modifying $\delta$ as a function of overall particle composition. (E) and (F) show the calculated $\sigma$ values for the systems shown in (C) and (D).





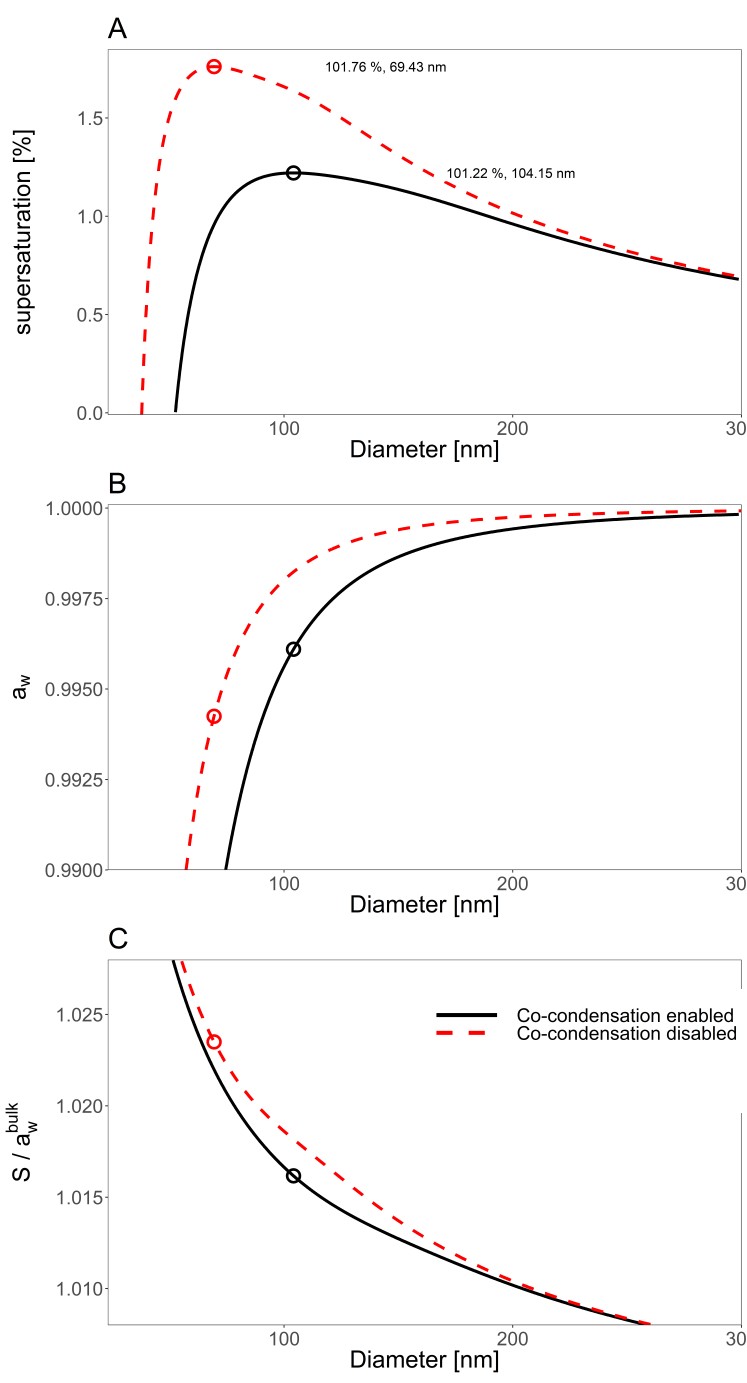

**Figure 7.** (A) The effect of including co-condensation of SVOC species for an aqueous 21-component system of isoprene-derived SOA (Rastak et al., 2017; Gervasi et al., 2020) with $D_{dry} = 23$ nm. (B) The bulk-phase water activity, corresponding to the contribution of the Raoult effect to the Köhler curves shown in (A). Panel (C) shows the Kelvin effect term's contribution to the curves in (A). For the co-condensation-disabled case, the particle's water-free composition was taken at RH = 0.1% and then only water was allowed to partition to and from particles with this dry composition. For a detailed description of the system components, see Table S1 in the SI.



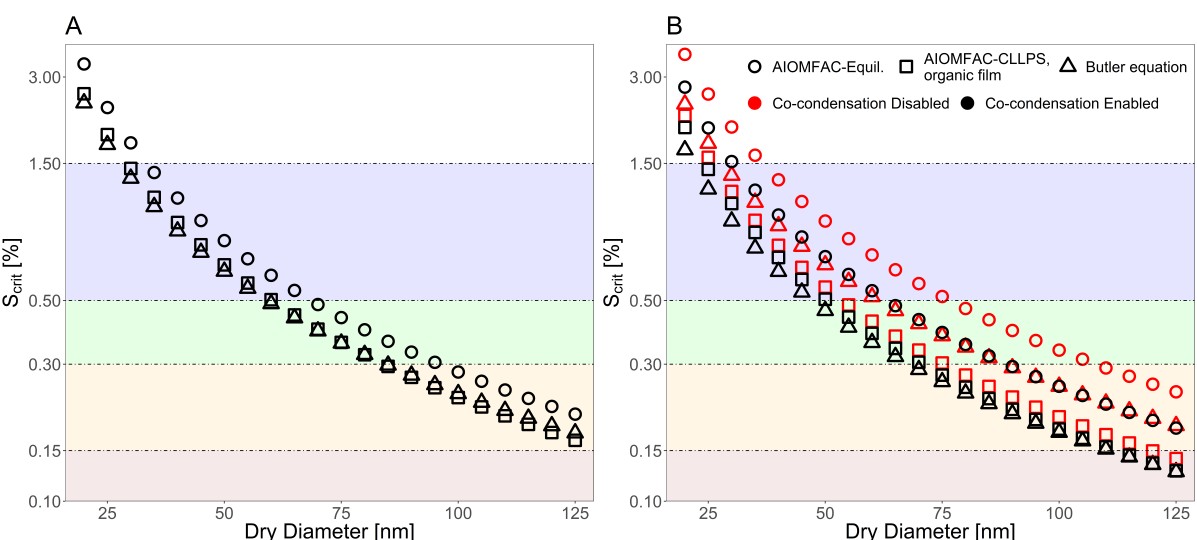

**Figure 8.** Critical supersaturations vs. dry diameter as predicted by the AIOMFAC-Equil. model, the AIOMFAC-CLLPS with organic film model, and a model from this work (Eq. 24) for (A) water–suberic acid–ammonium sulfate particles and (B) water–isoprene-SOA particles. The horizontal shaded bands represent distinct regimes of maximum supersaturations encountered by aerosols in either marine (blue), clean continental (green), background (orange-yellow), or urban polluted (brown) conditions at cloud base, as reported by Pinsky et al. (2014).





**Table 1.** Partial molar areas ( $m^2\,mol^{-1}$ ) of different organic species calculated using a simplified geometric approach based on the values given in Topping et al. (2007) and the model developed by Goldsack and White (1983), as well as by Eq. (19) of this study for particle diameters of 5, 50, and 5000 nm and two distinct $\delta$ values (0.15 nm or 1.0 nm).

| Species | Previous Treatments | | This study | | | | | |
| | Goldsack & White (1983) | Simplified Geometric Approach | $D = 5$ nm | | $D = 50$ nm | | $D = 5000$ nm | |
| | | | $\delta = 0.15$ nm | $\delta = 1.0$ nm | $\delta = 0.15$ nm | $\delta = 1.0$ nm | $\delta = 0.15$ nm | $\delta = 1.0$ nm |
|---|---|---|---|---|---|---|---|---|
| Succinic Acid | $3.43 \times 10^6$ | $1.76 \times 10^5$ | $5.12 \times 10^5$ | $8.41 \times 10^4$ | $5.05 \times 10^5$ | $7.65 \times 10^4$ | $5.05 \times 10^5$ | $5.05 \times 10^5$ |
| Malonic Acid | $4.98 \times 10^6$ | $1.51 \times 10^5$ | $4.35 \times 10^5$ | $7.14 \times 10^4$ | $4.29 \times 10^5$ | $6.49 \times 10^4$ | $4.28 \times 10^5$ | $4.28 \times 10^5$ |
| Oxalic Acid | $2.56 \times 10^6$ | $1.31 \times 10^5$ | $3.21 \times 10^5$ | $5.26 \times 10^4$ | $3.16 \times 10^5$ | $4.79 \times 10^4$ | $3.16 \times 10^5$ | $3.16 \times 10^5$ |
| Glutaric Acid | $3.78 \times 10^6$ | $1.93 \times 10^5$ | $6.28 \times 10^5$ | $1.03 \times 10^5$ | $6.19 \times 10^5$ | $9.37 \times 10^4$ | $6.19 \times 10^5$ | $6.19 \times 10^5$ |
| Citric Acid | $4.25 \times 10^6$ | $2.16 \times 10^5$ | $8.43 \times 10^5$ | $1.38 \times 10^5$ | $8.32 \times 10^5$ | $1.26 \times 10^5$ | $8.31 \times 10^5$ | $8.31 \times 10^5$ |
| Malic Acid | $3.59 \times 10^6$ | $1.88 \times 10^5$ | $5.67 \times 10^5$ | $9.31 \times 10^4$ | $5.59 \times 10^5$ | $8.46 \times 10^4$ | $5.58 \times 10^5$ | $5.58 \times 10^5$ |
| Maleic Acid | $3.34 \times 10^6$ | $1.69 \times 10^5$ | $4.94 \times 10^5$ | $8.11 \times 10^4$ | $4.87 \times 10^5$ | $7.37 \times 10^4$ | $4.87 \times 10^5$ | $4.87 \times 10^5$ |
| Adipic Acid | $4.25 \times 10^6$ | $2.29 \times 10^4$ | $7.27 \times 10^5$ | $1.19 \times 10^5$ | $7.17 \times 10^5$ | $1.09 \times 10^5$ | $7.16 \times 10^5$ | $7.16 \times 10^5$ |
| Fulvic Acid | $1.04 \times 10^7$ | $4.83 \times 10^5$ | $1.17 \times 10^6$ | $1.91 \times 10^5$ | $1.15 \times 10^6$ | $1.74 \times 10^5$ | $1.15 \times 10^6$ | $1.15 \times 10^6$ |
| Levoglucosan | $4.21 \times 10^6$ | $2.14 \times 10^5$ | $6.44 \times 10^5$ | $1.06 \times 10^5$ | $6.35 \times 10^5$ | $9.61 \times 10^5$ | $6.34 \times 10^5$ | $6.34 \times 10^5$ |
| Pinic Acid | $4.99 \times 10^6$ | $2.69 \times 10^5$ | $1.05 \times 10^6$ | $1.72 \times 10^5$ | $1.04 \times 10^5$ | $1.57 \times 10^5$ | $1.03 \times 10^6$ | $1.03 \times 10^6$ |





**Table 2.** Critical wet diameter and supersaturations for the Köhler curves shown in Figure 6.

| Model type | $\delta$ [nm] | $\mathscr{A}_i^{\circ}$ | $D_{dry} = 150$ nm | | $D_{dry} = 40$ nm | |
| | | | $D_{crit}$ [nm] | $SS_{crit}$ [%] | $D_{crit}$ [nm] | $SS_{crit}$ [%] |
| --- | --- | --- | --- | --- | --- | --- |
| Eq. (24) | $\delta = 0.3$ nm | $\mathscr{A}_i^{\circ} = \mathscr{A}_i$ | 1127.0 | 0.13 | 216.8 | 0.84 |
| Eq. (24) | $\delta = 0.15$ nm | $\mathscr{A}_i^{\circ} = \mathscr{A}_i$ | 1153.4 | 0.09 | 296.3 | 0.64 |
| Eq. (40) | $\delta = f(x_i^{surf}, ..., x_k^{surf})$ | $\mathscr{A}_i^{\circ} \neq \mathscr{A}_i$ | 1505.8 | 0.12 | 208.8 | 0.85 |
| Eq. (24) | $\delta = f(x_i^{surf}, ..., x_k^{surf})$ | $\mathscr{A}_i^{\circ} = \mathscr{A}_i$ | 1118.3 | 0.14 | 212.29 | 0.81 |
| Eq. (40) | $\delta = 0.3$ nm | $\mathscr{A}_i^{\circ} \neq \mathscr{A}_i$ | 1540.4 | 0.11 | 190.9 | 0.89 |
| AIOMFAC-Equil | $\delta = 0.3$ nm | – | 929.2 | 0.15 | 124.1 | 1.1 |
| AIOMFAC-CLLPS, organic film mode | $\delta = 0.3$ nm | – | 1713.0 | 0.09 | 108.1 | 0.62 |