# Peer review of "A thermodynamic framework for bulk-surface partitioning in finite-volume mixed organic-inorganic aerosol particles and cloud droplets"

_EGUsphere, 2023_

## Author Comment (AC3)

We would like to thank the referee for their helpful comments and feedback. As such, we have included responses to their major comment below:

*Schmedding and Zuend treat the interfacial region (boundary between the liquid and vapor phase) as a separate phase following the approach of Guggenheim. This leads to a significant simplification, since the chemical potential in the interfacial region is assumed to be equal to that in the other phases.*

    *Rowlinson and Widom point out that "we can not measure or define unambiguously and independently the thermodynamic properties of the surface phase". They go on to say "We can evade the difficulty only by defining the properties of the*
*surface phase as differences between those of the whole system and those of the two phases. They point out that questions about local thermodynamic functions (density, chemical potential) in the interface region are best answered by the methods of statistical mechanics.*

    *Chapter 5 of their book includes calculations of the local thermodynamic functions based on statistical mechanics for a number of models including hard spheres, lattice gas model, and penetrable sphere model for one and two component*
*systems. It would greatly increase the impact of this study if the authors included a comparison with results given by Rowlinson and Widom. One possibility is a comparison of the modified chemical potential based on statistical mechanics (Eq. (4–68)) versus the value obtained using the Guggenheim approach for the two component lattice gas.*

We largely agree with the cited statement from Rowlinson and Widom regarding difficulties in unambiguously defining prop-
erties of a surface phase. However, based on a few assumptions, one can represent the thermodynamic properties of a droplet with a surface phase in a thermodynamically sound and self-consistent manner, which is what we propose in our work. The key assumption made herein is that the there are neither spatially well-resolved density nor concentration gradients between the bulk liquid and gas phases; rather, we assume that the surface phase has a liquid-like density and potentially distinct composition compared to the droplet bulk. The sensitivity of bulk–surface partitioning of different species to the density is at least
partially captured by exploring changes to the values of $\delta$ and thus $\mathscr{A}_i$, as discussed in the context of Eqs. (14–19) of Section 2.

    Indeed, through the definition of the local chemical potential of the surface phase (Eqs. 10–13), and the subsequent derivation of the Butler equation (Eqs. 21–24), it is shown that the local chemical potential is in fact defined in relation to the bulk liquid phase beneath it and, via the surface tension, experiences influence from the contact to the gas phase. If a finite depth layer were to be placed entirely within a homogeneously mixed liquid bulk solution, then a trivial solution to Eq. (24) would be
found. This suggests that these molecules experience no additional penalty for their presence completely within a bulk phase. On the other hand, molecules at the surface must experience a form of an energetic penalty due to the geometric constraints on the directions in which they can interact with other molecules. In order to account for possible differences in the activity coefficients of species in the bulk and surface phases, we introduced the single exponential scaling factor $t$, which could be applied to all surface activity coefficients as mentioned in Section 2.3 and explored in greater detail in Figs. S3–S6.

As outlined further below (under Related Manuscript Changes), the thermodynamic model we apply to all phases, including the surface phase, is directly rooted in an advanced statistical mechanics model of local composition effects on nonideal mixing. As such, while a comparison to a variety of other possible statistical mechanics-derived models for surface chemical potentials for one- and two-component systems might be of some interest, our intent is to consistently use the same comprehensive activity coefficient model (AIOMFAC) for all phases – a model that is applicable to any number of components. A
detailed comparison exercise is beyond the intent and scope of this study.

**Related Manuscript Changes**

    The following text has been added to a revised version of Section 3 of the manuscript to clarify the statistical mechanics
underpinnings of AIOMFAC:

AIOMFAC can be considered as a major extension of the UNIversal quasichemical Functional Groups Activity Coefficients (UNIFAC) model by Fredenslund et al. (1975), since AIOMFAC includes the treatment of aqueous electrolytes and interactions between ions and organic molecules. UNIFAC itself is a group-contribution model derived from the UNIversal QUAsi-Chemical (UNIQUAC) theory of liquid mixtures developed by Abrams and Prausnitz (1975). The UNIQUAC theory and model is a local composition model for mixtures of non-electrolyte components that generalizes the original quasi-chemical theory of Guggenheim (1952) to mixtures consisting of molecules of various shapes and sizes (while Guggenheim's quasi-chemical lattice model was restricted to spherical molecules of approximately equal sizes). The development of the UNIQUAC model, the UNIFAC group-contribution version, and subsequently extended variants like AIOMFAC, are therefore all deeply rooted in an advanced statistical mechanics treatment of mixing and interactions amongst different molecules in solution based on the local composition principle and are more rigorous than Guggenheim's two-component lattice gas model.

An explanation of a statistical mechanics approach was also included in the revised version of Section 4.1 on the theoretical implications of this study:

An alternative approach to using the framework laid out in this work is to use other statistical mechanics models to predict the surface tension as a function of bulk solution composition. A simplified statistical mechanics-based approach for surface tension predictions was developed by Wexler and Dutcher (2013), which relies on a single physically constrained fitted parameter, $r$, which represents the average number of water molecules displaced by a solute molecule at the surface. As shown in Figure S6, this model had a root mean square error of $2.90 \ \mathrm{mJm^{-2}}$ in comparison to measurements of surface tension for a (macroscopic) binary water–ethanol system (Ernst et al., 1935). In comparison, the model developed in this work has a root mean square error of $2.930 \ \mathrm{mJm^{-2}}$, when using our default assumption that the thickness of the interface is $\delta = 0.3$ nm. The fitted value of $r$ as reported by Wexler and Dutcher (2013) is 3.00. If the number of water molecules displaced by an ethanol molecule at the surface of a droplet is assumed to be determined based on the respective values of $\mathscr{A}_{water}$ and $\mathscr{A}_{ethanol}$, then the number of water molecules displaced by an ethanol molecule in the surface phase is 3.02. Despite both models being in good agreement in the macroscopic case, it is important to note that their statistical mechanical model does not directly account for bulk-phase depletion in volume-constrained systems. For a comparison of various other frameworks for estimating the surface tension of liquid solutions and/or atmospheric aerosol particles, we refer the reader to the recent work by Kleinheins et al. (2023) and Vepsäläinen et al. (2022, 2023).

In addition the following minor changes have been implemented based on the reviewer's feedback.

***Equation 1. - multiplied by 100% is a typo. (e.g, one multiplies the value by 100, such that the value is a percentage)***
Equation 1 has been updated to reflect a calculation of the saturation ratio, with the following text added to lines 93 and 94 :

Note that values of $S$ are often reported as percentages, in which case the value of Eq. (1) is multiplied by $100\%$.

***Equation 23 - is sigma missing a subscript? this sigma is somewhat ambiguous.***
Equation 23 has been updated to clarify that $\sigma$ refers to $\sigma_i$, the computed surface tension contribution of species $i$ in solution.
$$\mu_i^s = \mu_i^{\circ,s} + RT\ln(a_i^s) + \sigma_i^\circ \mathscr{A}_i^\circ - \sigma_i \mathscr{A}_i.$$
Note that under equilibrium conditions for all $k$ species in solution $\sigma_1 = \sigma_2 = ... = \sigma_k = \sigma$.

**References**

Abrams, D. S. and Prausnitz, J. M.: Statistical thermodynamics of liquid mixtures: A new expression for the excess Gibbs energy of partly or completely miscible systems, AIChE Journal, 21, 116–128, https://doi.org/https://doi.org/10.1002/aic.690210115, 1975.

Ernst, R. C., Watkins, C. H., and Ruwe, H.: The Physical Properties of the Ternary System Ethyl Alcohol–Glycerin–Water., The Journal of Physical Chemistry, 40, 627–635, 1935.

Fredenslund, A., Jones, R. L., and Prausnitz, J. M.: Group-contribution estimation of activity coefficients in nonideal liquid mixtures, AIChE Journal, 21, 1086–1099, https://doi.org/https://doi.org/10.1002/aic.690210607, 1975.

Guggenheim, E. A. (Edward Armand), .-.: Mixtures; the theory of the equilibrium properties of some simple classes of mixtures, solutions and alloys., The International series of monographs on physics, Clarendon Press, Oxford, 1952.

Kleinheins, J., Shardt, N., El Haber, M., Ferronato, C., Nozière, B., Peter, T., and Marcolli, C.: Surface tension models for binary aqueous solutions: a review and intercomparison, Phys. Chem. Chem. Phys., 25, 11 055–11 074, https://doi.org/10.1039/D3CP00322A, 2023.

Vepsäläinen, S., Calderón, S. M., Malila, J., and Prisle, N. L.: Comparison of six approaches to predicting droplet activation of surface active aerosol – Part 1: moderately surface active organics, Atmospheric Chemistry and Physics, 22, 2669–2687, https://doi.org/10.5194/acp-22-2669-2022, 2022.

Vepsäläinen, S., Calderón, S. M., and Prisle, N. L.: Comparison of six approaches to predicting droplet activation of surface active aerosol – Part 2: strong surfactants, EGUsphere, 2023, 1–23, https://doi.org/10.5194/egusphere-2022-1188, 2023.

Wexler, A. S. and Dutcher, C. S.: Statistical Mechanics of Multilayer Sorption: Surface Tension, The Journal of Physical Chemistry Letters, 4, 1723–1726, https://doi.org/10.1021/jz400725p, pMID: 26282984, 2013.